# Wormholes from averaging over states

**Ben Freivogel[1,2]\*, Dora Nikolakopoulou[1]† and Antonio F. Rotundo[1]‡**

**1** Institute for Theoretical Physics Amsterdam
and Delta Institute for Theoretical Physics
**2** GRAPPA, University of Amsterdam, Science Park 904,
1090 GL Amsterdam, the Netherlands

\* benfreivogel@gmail.com , † t.nikolakopoulou@uva.nl , ‡ af.rotundo@gmail.com

## Abstract

An important question about black holes is to what extent a typical pure state differs from the ensemble average. We show that this question can be answered within semi-classical gravity. We focus on the *quantum deviation*, which measures the fluctuations in the expectation value of an operator in an ensemble of pure states. For a large class of ensembles and observables, these fluctuations are calculated by a correlation function in the eternal black hole background, which can be reliably calculated within semi-classical gravity. This implements the idea of [1] that wormholes can arise from averages over states rather than theories. As an application, we calculate the size of the long-time correlation function $\langle A(t)A(0)\rangle$.

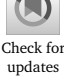

# 1   Introduction and main results

Recent progress has shown that semiclassical gravity has access to information about the spectrum of the underlying microscopic theory which was previously thought to be beyond its reach. Two notable examples are the ramp in the spectral form factor [2] and the Page curve [3,4] (see [5] for a review). A common aspect of these works is the inclusion of wormhole geometries in the gravitational path integral.

These wormholes join boundaries which are otherwise disconnected and induce correlations between them. This is puzzling from a holographic perspective because on the non-gravitational side of the duality there is no coupling between the boundaries and hence no correlation. This inconsistency is particularly sharp if we consider the product of partition functions, as in the spectral form factor. On the boundary theory we are simply multiplying two functions. However, on the gravitational side, wormhole geometries lead to an answer which does not factorize [6].

A simple way to resolve this inconsistency is to assume that semiclassical gravity is computing some sort of average [3,6–10]. The correlations given by the wormholes correspond with the ones generated by the averaging. This is indeed the case in the examples considered in [3,6], as they are based on JT gravity, which is known to be dual to an ensemble of theories [11]. However, in higher dimensions there seems to be a problem. On the one hand there is no clear reason why wormhole geometries, analogous to the ones that contribute in JT gravity, should be discarded. On the other hand the boundary theory is believed to be unique. This leads to the question of what sort of averaging, if any, gravity is performing.

One interesting resolution of this problem was proposed in [1]. See also [12–16] for related works. The idea is that, within the semiclassical approximation, we can only probe a gravitational system through operators that cannot resolve the detailed structure of the Hilbert space. Therefore, semiclassical gravity effectively computes averages over ensembles of typical states. The main advantage of this approach is that the boundary theory is unique. Moreover, it provides a simple and intuitive explanation for the average. Notice that this type of reasoning is familiar in the context of ETH [17,18] and the typicality approach to the foundation of statistical mechanics [19–21].

In this paper we implement this proposal. The authors of [1] consider a large class of observables and write down an effective field theory that encodes their ensemble averages and higher moments. We focus on one particular example of these averaged quantities which we call the *quantum deviation*, defined by

$$\Delta_{\mathcal{O}}^2 \equiv \left[\!\!\left[ \left| \text{Tr}(\mathcal{O}\rho) \right|^2 \right]\!\!\right] - \left| \left[\!\!\left[ \text{Tr}(\mathcal{O}\rho) \right]\!\!\right] \right|^2. \tag{1}$$

Here $[\![\ldots]\!]$ indicates averaging over an ensemble of states, which can be either pure or mixed. Notice that in the equation above, $\rho$ is a state drawn from the ensemble, not to be confused

with the ensemble itself. In particular we want to understand whether this quantity can be reliably computed in semiclassical gravity for different choices of ensembles.

The quantum deviation is a very different quantity than the usual variance of a quantum operator. The variance, $\langle \mathcal{O}^2 \rangle - \langle \mathcal{O} \rangle^2$, gives a measure for the spread in measurement outcomes in any given quantum state. On the other hand, the quantum deviation captures fluctuations in the expectation value in some ensemble of states. While the usual variance can be calculated given the density matrix of the system, the quantum deviation is only defined once we have specified a probability distribution over quantum states. Therefore, $\Delta_O$ is suitable for determining, for example, whether typical pure state black holes have structure at the horizon that disappears after averaging over states.

Another reason why the quantum deviation is an interesting quantity to study is that, if we assume that semiclassical gravity only computes averages, it tells us how far we can trust the semiclassical approximation. We should not trust the average value of an observable when it is comparable to $\Delta_O$.

Our main results are

- When averaging over a microcanonical ensemble of pure states, as in [1], the quantum deviation is given by a correlation function in the 'microcanonical double' state,[1]

$$\Delta_O^2 = e^{-S} \langle \mathrm{MCD} | \, \mathcal{O}_L \mathcal{O}_R \, | \mathrm{MCD} \rangle \,. \tag{2}$$

  The microcanonical double state $|\mathrm{MCD}\rangle$ is the microcanonical analog of the thermofield double state. This result relies only on having a large number of states in the microcanonical window, and does not require us to assume ETH.

- In the regime of interest, the state $|\mathrm{MCD}\rangle$ does *not* have a simple gravity dual, so this quantity cannot be computed using semiclassical gravity.

- We generalize (2) to a large class of ensembles. In particular if we average over a 'canonical' ensemble of pure states with inverse temperature $\beta$, the quantum deviation is given by

$$\Delta_O^2 = \frac{Z(2\beta)}{Z(\beta)^2} \Big[ \big\langle \mathrm{TFD}_{2\beta} \big| \, \mathcal{O}_L \mathcal{O}_R \, \big| \mathrm{TFD}_{2\beta} \big\rangle + O(e^{-S}) \Big]. \tag{3}$$

- In holographic theories, for a broad class of operators $\mathcal{O}$, the quantum deviation $\Delta_O$ in this 'canonical' ensemble of pure states can be reliably computed in semiclassical gravity via a simple wormhole saddle, the eternal black hole.

- Semiclassical gravity does *not* always calculate the mean value $[\![\langle \psi | \mathcal{O} | \psi \rangle]\!]$ correctly. Specifically, in calculating long-time correlation functions, $\mathcal{O} = A(t)A(0)$, semiclassical gravity does *not* correctly calculate the average over states $[\![\langle \psi | \mathcal{O} | \psi \rangle]\!]$ for sufficiently large $t$. However, it *does* reliably calculate the quantum deviation of this quantity. In the regime where the deviation is small compared to the average, semiclassical gravity does compute the average correctly.

Taken together, these results indicate that we can calculate, within semiclassical gravity, the size of quantum gravity effects on a wide class of correlation functions. This is surprising because $\Delta_O$ is exponentially suppressed in the entropy. Therefore, one might expect that this quantity is not accessible to semiclassical gravity at all. We calculate the average size of these effects in a class of pure states, but averaging over theories is not needed.

Finally, (3) provides an explicit example of a connected geometry emerging from an average over states, and it strengthens the proposal that averages over states are sufficient to

---

[1]For simplicity we assume that the disconnected part of the correlator is zero in this introductory section.

resolve the factorization problem. However, notice that we don't consider products of partition functions, where, as we mentioned at the beginning of this section, the factorization problem is particularly sharp. In particular, the wormholes we find are Lorentzian, and not Euclidean as in the calculation of products of partition functions.

## 2 Quantum deviation

In this section, we define the quantum deviation more carefully. In particular we show that it can be computed as an expectation value in a double copy of the theory.

In standard quantum mechanics, every aspect of the quantum state is encoded in the density matrix; the results of all possible measurements done on the system can be calculated from the density matrix.

Here, motivated by the gravity dual, we want to allow more elaborate experiments. Imagine we are given an ensemble of quantum states $\rho$ with associated probabilities $P(\rho)$. We are allowed to not only do experiments on single copies of the system, but also to do experiments on multiple copies at once. Given such a quantum ensemble, we define the quantum deviation, $\Delta_O$, by

$$\Delta_O^2 \equiv \left[\!\left[ \left| \text{Tr}(\mathcal{O}\rho) \right|^2 \right]\!\right] - \left| \left[\!\left[ \text{Tr}(\mathcal{O}\rho) \right]\!\right] \right|^2 . \tag{4}$$

Here the double brackets indicate the average over $\rho$, $[\![\cdot]\!] \equiv \sum_\rho P(\rho)\cdot$, depending on the ensemble considered, the sum over $\rho$ should be replaced by an integral. The quantum deviation for ensembles of pure states can be obtained by setting $\rho = |\psi\rangle\langle\psi|$. Since in the rest of the paper we will mostly consider ensemble of pure states, it is convenient to write it down explicitly:

$$\Delta_O^2 = \left[\!\left[ \left| \langle\psi| \mathcal{O} |\psi\rangle \right|^2 \right]\!\right] - \left| \left[\!\left[ \langle\psi| \mathcal{O} |\psi\rangle \right]\!\right] \right|^2 . \tag{5}$$

This type of quantum ensemble has been considered before, and we have not attained a comprehensive knowledge of the literature. We are aware of [21–23].

To better illustrate the information carried by the quantum deviation, it is helpful to study a simple example. We consider two different ensembles for a spin $\frac{1}{2}$ particle. In distribution $A$ there is equal probability for all pure states,

$$A \equiv \left\{ \left( |\hat{n}\rangle , p_{\hat{n}} = \frac{1}{4\pi} \right) \right\}_{\hat{n} \in S_2} , \tag{6}$$

where $\hat{n}$ is a point on the Bloch sphere and

$$|\hat{n}\rangle = \cos\frac{\theta}{2} |\uparrow\rangle + e^{i\phi} \sin\frac{\theta}{2} |\downarrow\rangle . \tag{7}$$

In distribution $B$ the particle is in the state $|\uparrow\rangle$ with probability $1/2$ and in the state $|\downarrow\rangle$ with probability $1/2$,

$$B \equiv \left\{ \left( |\uparrow\rangle , p_\uparrow = \frac{1}{2} \right), \left( |\downarrow\rangle , p_\downarrow = \frac{1}{2} \right) \right\} . \tag{8}$$

Let $\mathcal{O} = \sigma_z$, with $\sigma_z |\uparrow\rangle = 1$ and $\sigma_z |\downarrow\rangle = -1$. The averaged expectation values of this operator in the two ensembles can be easily computed. For ensemble $A$ we have

$$A: \quad [\![\langle\sigma_z\rangle]\!] = \frac{1}{4\pi} \int d\Omega_2 \, \langle\hat{n}| \sigma_z |\hat{n}\rangle = 0 , \tag{9}$$

where $d\Omega_2$ is the volume element on the Bloch sphere. For ensemble $B$ we have

$$B: \quad [\![\langle\sigma_z\rangle]\!] = \frac{1}{2} \left( \langle\uparrow| \sigma_z |\uparrow\rangle + \langle\downarrow| \sigma_z |\downarrow\rangle \right) = 0 . \tag{10}$$

We see that the averaged expectation value of $\sigma_z$ cannot distinguish between the two ensembles. The quantum deviation on the other hand can. We have

$$A: \quad \Delta^2_{\sigma_z} = \frac{1}{4\pi}\int d\Omega_2 \, \langle \hat{n}|\, \sigma_z\,|\hat{n}\rangle^2 = \frac{1}{3}\,, \tag{11}$$

while

$$B: \quad \Delta^2_{\sigma_z} = \frac{1}{2}\big(\langle\uparrow|\,\sigma_z\,|\uparrow\rangle^2 + \langle\downarrow|\,\sigma_z\,|\downarrow\rangle^2\big) = 1\,. \tag{12}$$

One might worry that the averaged expectation value of some other operator could distinguish between the two ensembles. To see that this is not the case it is convenient to rewrite the averaged expectation value as

$$[\![\mathrm{Tr}(\rho\,\mathcal{O})]\!] = \mathrm{Tr}(\rho_1 \mathcal{O})\,, \tag{13}$$

where we have defined $\rho_1 \equiv [\![\,\rho\,]\!]$. Going back to the spin example it's easy to show that for both ensembles $A$ and $B$

$$A,B: \quad \rho_1 = \frac{1}{2}\big(|\uparrow\rangle\langle\uparrow| + |\downarrow\rangle\langle\downarrow|\big)\,, \tag{14}$$

so the averaged expectation value of *any* operator is the same between the two ensembles. The main advantage of the rewriting above is that we need to perform the average once, to compute the averaged state $\rho_1$. Averaged expectation values then correspond to usual expectation values in this averaged state.

We can rewrite also the quantum deviation in terms of an averaged state, at the cost of considering a doubled theory. We have

$$\left[\!\left[\,\big|\mathrm{Tr}(\mathcal{O}\rho)\big|^2\,\right]\!\right] = \mathrm{Tr}\big(\rho_2\,\mathcal{O}^\dagger \otimes \mathcal{O}\big)\,, \tag{15}$$

where we have defined a normalized density matrix in a doubled Hilbert space,

$$\rho_2 \equiv [\![\,\rho \otimes \rho\,]\!]\,. \tag{16}$$

The quantum deviation is then given by

$$\Delta^2_{\mathcal{O}} = \mathrm{Tr}\big(\rho_2\,\mathcal{O}^\dagger \otimes \mathcal{O}\big) - \big|\mathrm{Tr}(\rho_1 \mathcal{O})\big|^2\,. \tag{17}$$

It's easy to see that tracing out either side in $\rho_2$ one obtains $\rho_1$. Therefore, the negative term in the expression above is simply subtracting the disconnected piece of the correlator, so the final result is simply a connected 2-point function in the doubled theory,

$$\Delta^2_{\mathcal{O}} = \mathrm{Tr}\big(\rho_2\big(\mathcal{O}^\dagger - \langle\mathcal{O}^\dagger\rangle\big) \otimes \big(\mathcal{O} - \langle\mathcal{O}\rangle\big)\big)\,, \tag{18}$$

where $\langle\mathcal{O}\rangle = \mathrm{Tr}(\rho_1 \mathcal{O})$. Going back again to the spin example, we can check that the double state $\rho_2$ is different in the two ensembles. For the ensemble A we have

$$A: \quad \rho_2 = \frac{1}{4\pi}\int d\Omega_2\, |n\,n\rangle\langle n\,n| = \frac{1}{3}\big(|\downarrow\downarrow\rangle\langle\downarrow\downarrow| + |\uparrow\uparrow\rangle\langle\uparrow\uparrow| + |\psi_+\rangle\langle\psi_+|\big)\,, \tag{19}$$

where we have defined $|\psi_+\rangle \equiv \big(|\uparrow\downarrow\rangle + |\downarrow\uparrow\rangle\big)/\sqrt{2}$. For ensemble B we have

$$B: \quad \rho_2 = \frac{1}{2}\big(|\downarrow\downarrow\rangle\langle\downarrow\downarrow| + |\uparrow\uparrow\rangle\langle\uparrow\uparrow|\big)\,. \tag{20}$$

We conclude that the distributions A and B lead to the same $\rho_1$ but different $\rho_2$'s. We can distinguish A from B by computing the quantum deviation for any operator $\mathcal{O}$ that is sensitive to this difference, such as $\sigma_z$.

We have shown that the quantum deviation is calculated by a correlation function in some state in the doubled theory. The non-trivial work we do in the rest of the article is to identify in what situations this state has a nice gravity dual.

However, first, we point out an interesting fact about the quantum deviation that further motivates its study. We show that the quantum deviation is sensitive to the averaged purity of the states in the ensemble considered.

## 2.1 Purity

We want to show that the quantum deviation is sensitive to whether we have an ensemble of pure states or mixed states.

We begin by inserting complete sets of energy eigenstates in (4)

$$\Delta_O^2 = \sum_{n_1 m_1} \sum_{n_2 m_2} \mathcal{O}_{m_1 n_1}^* \mathcal{O}_{n_2 m_2} \Big( [\![\rho_{m_1 n_1} \rho_{m_2 n_2}]\!] - [\![\rho_{m_1 n_1}]\!][\![\rho_{m_2 n_2}]\!] \Big). \tag{21}$$

We can't compute the quantum deviation without knowing something about the ensemble we are averaging over in $[\![\cdot]\!]$. However, it turns out that we can make some progress if we assume that the operator $\mathcal{O}$ obeys ETH [17, 18, 24],

$$\mathcal{O}_{nm} = \mathcal{O}(\bar{E})\delta_{nm} + e^{-S(\bar{E})/2} f(\bar{E}, \omega) R_{nm}. \tag{22}$$

Here $\bar{E} = (E_n + E_m)/2$, $\omega = E_m - E_n$, $\mathcal{O}(\bar{E})$ is the microcanonical expectation value of $\mathcal{O}$ at energy $\bar{E}$ and $R_{nm}$ is a random number with zero mean and unit variance. Using this we find

$$\begin{aligned}
\Delta_O^2 &= \sum_{nm} \mathcal{O}(E_n)\mathcal{O}(E_m)\Big( [\![\rho_{nn}\rho_{mm}]\!] - [\![\rho_{nn}]\!][\![\rho_{mm}]\!] \Big) \\
&+ \sum_{nm} e^{-S(\bar{E})} \big| f(\bar{E}, \omega) \big|^2 \Big( [\![|\rho_{nm}|^2]\!] - \big| [\![\rho_{nm}]\!] \big|^2 \Big).
\end{aligned} \tag{23}$$

Here to simplify the expressions we have replaced the random variables $R_{nm}$ and their products with their averages: $R_{nm} \to 0$ and $R_{m_1 n_1}^* R_{n_2 m_2} \to \delta_{m_1 n_2} \delta_{n_1 m_2}$.

The term in the first line can be simplified if we expand $\mathcal{O}(E)$ around the mean energy $E_* = \text{Tr}(\rho_1 H)$,

$$\mathcal{O}(E_n) = \mathcal{O}(E_*) + \mathcal{O}'(E_*)(E_n - E_*) + \frac{1}{2}\mathcal{O}''(E_*)(E_n - E_*)^2. \tag{24}$$

Plugging this in (23) and keeping only terms up to quadratic order in $(E - E_*)$ we find

$$\Delta_O^2 = \mathcal{O}'(E_*)^2 \Delta_H^2 + \sum_{nm} e^{-S(\bar{E})} \big| f(\bar{E}, \omega) \big|^2 \Big( [\![|\rho_{nm}|^2]\!] - \big| [\![\rho_{nm}]\!] \big|^2 \Big), \tag{25}$$

where $\Delta_H^2$ is the quantum deviation of the Hamiltonian. The first term is not very exciting; it depends on the operator only through its microcanonical expectation value at energy $E_*$. The second term instead can be related to the averaged purity in the ensemble, as we now show.

To estimate the second term, we approximate $S(\bar{E})$ and $\big| f(\bar{E}, \omega) \big|^2$ with constants and take them out of the sums. We find[2]

$$\Delta_O^2 \approx \mathcal{O}'(E_*)^2 \Delta_H^2 + e^{-S} |f|^2 \Big( [\![\text{Tr}\,\rho^2]\!] - \text{Tr}[\![\rho]\!]^2 \Big). \tag{26}$$

---

[2]We thank the anonymous referee for pointing out this simple way of writing the last term in (26).

The term in parentheses is the difference between the average purity of the states in the ensemble and the purity of the average state. For typical ensembles, $[\![\mathrm{Tr}\,\rho^2]\!] \gg \mathrm{Tr}[\![\rho]\!]^2$. We can see this as follows. Let $|a_\rho\rangle$ be the basis in which the matrix $\rho$ is diagonal

$$\rho = \sum_{a_\rho} \rho_{aa} |a_\rho\rangle\langle a_\rho| \,. \tag{27}$$

Then we have

$$\mathrm{Tr}[\![\rho]\!]^2 = \sum_{\rho,\rho'} P(\rho)P(\rho') \sum_{a_\rho,b_{\rho'}} \rho_{aa}\rho'_{bb} \big|\langle a_\rho|b_{\rho'}\rangle\big|^2 \,. \tag{28}$$

The bases for different $\rho$ don't need being orthonormal; however, for typical ensembles we expect that, given two randomly drawn matrices $\rho$ and $\rho'$, the elements of their bases are approximately orthogonal, $\langle a_\rho|b_{\rho'}\rangle \approx \delta_{ab}\delta(\rho-\rho')$. Using this, we can simplify the expression above to

$$\mathrm{Tr}[\![\rho]\!]^2 \approx \sum_\rho P(\rho)^2 \mathrm{Tr}\,\rho^2 \,, \tag{29}$$

which is typically much smaller than $[\![\mathrm{Tr}\,\rho^2]\!] = \sum_\rho P(\rho)\mathrm{Tr}\,\rho^2$. We expect this to be true in typical ensembles, in which a large number of states have similar probabilities; however, we don't know how to precisely characterize these ensembles. It would be interesting to study this further.

For the moment, let's assume that the ensemble satisfies $[\![\mathrm{Tr}\,\rho^2]\!] \gg \mathrm{Tr}[\![\rho]\!]^2$. Eq. (26) reduces to

$$\Delta_O^2 \approx \mathcal{O}'(E_*)^2 \Delta_H^2 + e^{-S}|f|^2[\![\mathrm{Tr}\,\rho^2]\!] \,. \tag{30}$$

The quantum deviation is sensitive to the average purity of the states in the ensemble, as long as the first term is not too large. As we will see in the rest of the paper, the quantum deviation is exponentially suppressed in the entropy of the system, so the two terms, $\mathcal{O}'(E_*)^2 \Delta_H^2$ and $e^{-S}|f|^2[\![\mathrm{Tr}\,\rho^2]\!]$ are generically of the same order. Whether it is possible to neglect the first term depends on the details of both the ensemble and the operator considered. In the next section we will give a simple example in which this is the case. However, in the rest of the paper we will focus on ensembles of pure states. It would be interesting to study in more detail the quantum deviation for ensembles of mixed states.

## 3 Microcanonical ensemble

We calculate the quantum deviation for the 'microcanonical' ensemble of pure states of [1]. States are drawn with equal probability from some energy window, $I = [E - \Delta E, E]$, i.e. the states are Haar-random. The typical energy splitting between energy eigenstates in the window is given by $\Delta E/d$, where $d \approx e^{S(E)}$ is the number of energy eigenstates in the window. An observer can try to measure the energy of the system. However, as long as they can only act with simple operators their experiments cannot be sensitive to these exponentially small splittings. Therefore, it is natural to average over states drawn from some small window of energies.

As explained in the previous sections, to find the quantum deviation we compute the averaged states $\rho_1$ and $\rho_2$. Let $|\psi\rangle$ be the pure state we want to average over. Our strategy is to first expand in the energy eigenbasis, $\{|n\rangle\}$, so that

$$|\psi\rangle = \sum_n \psi_n |n\rangle \,. \tag{31}$$

Then we perform the average over the components of the state in this basis.

We leave the details of this average to Appendix A. To compute $\rho_1$ and $\rho_2$ we only need the expression for the 2-point function of $\psi_n$

$$[\![\psi_n \psi_m^*]\!] = \frac{1}{d} \delta_{nm} , \tag{32}$$

and the 4-point function

$$[\![\psi_{n_1} \psi_{n_2} \psi_{m_1}^* \psi_{m_2}^*]\!] = \frac{1}{d(d+1)} (\delta_{n_1 m_1} \delta_{n_2 m_2} + \delta_{n_1 m_2} \delta_{n_2 m_1}) . \tag{33}$$

To be more clear we compute $\rho_1$ explicitly,

$$\rho_1 = [\![|\psi\rangle\langle\psi|]\!] = \sum_{E_n, E_m \in I} [\![\psi_n \psi_m^*]\!] \, |n\rangle\langle m| = \frac{1}{d} \sum_{E_n \in I} |n\rangle\langle n| . \tag{34}$$

Here we have expanded in the energy eigenbasis and then used the equation for the 2-point function of $\psi_n$. The final result is the usual microcanonical ensemble from statistical mechanics. Following the same procedure for $\rho_2$, we find

$$\rho_2 = \frac{1}{d(d+1)} \sum_{nm} (|n\,m\rangle\langle n\,m| + |n\,m\rangle\langle m\,n|) . \tag{35}$$

These formulas were already known in the literature, see for example [20, 25].

The 'doubled' state $\rho_2$ can be rewritten in terms of the 'microcanonical double'

$$|\text{MCD}\rangle \equiv \frac{1}{\sqrt{d}} \sum_{E_n \in I} |\tilde{n}\rangle \, |n\rangle , \tag{36}$$

where $|\tilde{n}\rangle = \Theta |n\rangle$ and $\Theta$ is an anti-unitary operator such as CPT. We have

$$\rho_2 = \frac{d}{d+1} \rho_1 \otimes \rho_1 + \frac{1}{d+1} \left( \Theta_L^\dagger |\text{MCD}\rangle\langle\text{MCD}| \Theta_L \right)^{T_L} , \tag{37}$$

where the superscript $T_L$ indicates a partial transpose.[3]

It may seem perverse to rewrite the relatively simple density matrix $\rho_2$ in this way, but we will see below that this rewriting allows for a simple gravity dual, once we generalize to other ensembles in the next section.

Combining everything, the quantum deviation for a single operator is given, in this microcanonical example, by

$$\Delta_O^2 = \frac{1}{d+1} \langle\text{MCD}| (\mathcal{O}_L - \langle\mathcal{O}_L\rangle)(\mathcal{O}_R - \langle\mathcal{O}_R\rangle) |\text{MCD}\rangle . \tag{38}$$

Here we have defined $\mathcal{O}_L \equiv \Theta \mathcal{O} \Theta^\dagger \otimes \mathbb{I}$ and $\mathcal{O}_R \equiv \mathbb{I} \otimes \mathcal{O}$.

We learn that the quantum deviation is computed by the connected two-point function in a microcanonical version of the thermofield double. The thermofield double, at high enough temperatures, is dual to an eternal black hole in AdS. Intuitively one might think that the microcanonical double is dual to some microcanonical version of the eternal black hole. Below we show that in fact there is no semiclassical dual to this state.

---

[3]Notice that to define the partial transpose we need to pick a preferred basis. This is only necessary in this intermediate step; our final equation, (38), is basis independent.

**Mixed ensembles.** First however, we take a small detour. Using the formulas above for $\rho_1$ and $\rho_2$, we can check in an explicit example that the quantum deviation is sensitive to how mixed are the states in the ensemble considered. We consider mixed states of the form

$$\rho = \frac{1}{K} \sum_{i=1}^{K} |\psi_i\rangle\langle\psi_i| \,, \tag{39}$$

where $|\psi_i\rangle$ are i.i.d. random states, drawn from the same microcanonical ensemble. This means that to average over the states $\rho$ we need to average over each $|\psi_i\rangle$ separately. The averaged density matrix is now given by

$$[\![\rho]\!] = \frac{1}{K} \sum_{i=1}^{K} [\![|\psi_i\rangle\langle\psi_i|]\!] = \frac{1}{d} \sum_{E_n \in I} |n\rangle\langle n| \,, \tag{40}$$

where we have used the fact that the states are identically distributed and the expression for the 2-point function (32). The final result is independent of $K$: it is again the microcanonical ensemble we found above for the ensemble of pure states. Therefore, we cannot learn whether the states in the ensemble are mixed by computing averaged expectation values. For the double state we have

$$
\begin{aligned}
[\![\rho \otimes \rho]\!] &= \frac{1}{K^2} \sum_{i,j=1}^{K} [\![|\psi_i \psi_j\rangle\langle\psi_i \psi_j|]\!] \\
&= \frac{1}{K^2} \Big( \sum_{i \neq j} [\![|\psi_i\rangle\langle\psi_i|]\!] \otimes [\![|\psi_j\rangle\langle\psi_j|]\!] + \sum_i [\![|\psi_i \psi_i\rangle\langle\psi_i \psi_i|]\!] \Big) \\
&= \rho_1 \otimes \rho_1 + \frac{1}{K} \frac{1}{d+1} \sum_{nm} \Big( \frac{1}{d} |n\,m\rangle\langle m\,n| - \rho_1 \otimes \rho_1 \Big),
\end{aligned}
\tag{41}
$$

where in going to the second line we have used that the states are independently distributed and in going to the last line we have used the expression for the 4-point function (33). From this expression it's simple to compute the quantum deviation

$$\Delta_O^2 = \frac{1}{K} \frac{1}{d+1} \langle \text{MCD}| (\mathcal{O}_L - \langle\mathcal{O}_L\rangle)(\mathcal{O}_R - \langle\mathcal{O}_R\rangle) |\text{MCD}\rangle \,. \tag{42}$$

Comparing with what we found above for the ensemble of pure states, eq. (38), we see that the quantum deviation for the ensemble of mixed states is suppressed by a factor $K$. A similar calculation shows that the average purity in this ensemble is given by

$$[\![\text{Tr}\,\rho^2]\!] = \frac{1}{K} + \frac{1}{d} - \frac{1}{Kd} \,. \tag{43}$$

We see that for $K \ll d$ the average purity is indeed given by $1/K$.[4] Notice that this average purity was already computed in in [1].

## 3.1 No simple gravity dual

Following [1], we now specialize to a holographic d-dimensional CFT, defined on a sphere of radius $R$. The usual thermofield double, at sufficiently high temperatures, is dual to the two-sided eternal black hole. As is familiar from statistical mechanics the microcanonical and

---

[4]For large values of $K$ some of the assumptions we made in sec. 2.1 break down, so we don't expect to find an agreement. For example $\langle a_\rho | b_{\rho'} \rangle \approx \delta_{ab} \delta(\rho - \rho')$ is not a good approximation anymore.

canonical ensembles are equivalent in the thermodynamic limit. So one might hope that the microcanonical double is dual to the same geometry. We now show that this is not the case.

First, we briefly recall how the equivalence works in statistical mechanics. Microcanonical and canonical expectation values are the same if we match the temperature of the canonical ensemble with the mean energy in the microcanonical ensemble according to $\beta = S'(\langle E \rangle)$. Fluctuations scale differently in the two ensembles, but this difference vanishes in the thermodynamic limit. In the holographic dual, we consider large black holes, i.e. $r_h \gg \ell_A$, which dominate both the canonical and microcanonical ensembles. To leading order in $G_N$, there is no difference between these two ensembles, and we can replace $\rho_{mc}$ with $\rho_\beta$. So we can compute the microcanonical average of $\langle O \rangle$, by inserting the operator in a black hole background.

To compute the quantum deviation we need to consider also the expectation value of $\mathcal{O}_L \mathcal{O}_R$ in the microcanonical double. We now want to estimate the difference between the expectation values of operators in this state and in the thermofield double.[5] To do this we expand in the energy eigenbasis

$$\langle \text{MCD}| \mathcal{O}_L \mathcal{O}_R |\text{MCD}\rangle = \frac{1}{d} \sum_{E_n, E_m \in I} |\mathcal{O}_{nm}|^2, \tag{44}$$

and assume that the operator $\mathcal{O}$ obeys ETH, see (22). The diagonal terms in the sum, $n = m$, give a term approximately equal to the microcanonical expectation value squared, $[\text{Tr}(\mathcal{O}\rho_{mc})]^2$. The same terms in the thermofield double would give the square of the canonical expectation value $[\text{Tr}(\mathcal{O}\rho_\beta)]^2$. From what we have said above we know that these terms agree if we pick the right temperature, so we can focus on the off-diagonal terms. This corresponds to setting $\mathcal{O}(\bar{E}) = 0$ in (22). Keeping this in mind we have

$$\langle \text{MCD}| \mathcal{O}_L \mathcal{O}_R |\text{MCD}\rangle = \frac{1}{d} \sum_{E_n, E_m \in I} e^{-S(\bar{E})} \left| f(\bar{E}, \omega) \right|^2, \tag{45}$$

where we have approximated $|R_{nm}|^2$ with 1.

To proceed further we assume that the function $f(\bar{E}, \omega)$ is approximately equal to a constant $f$ in an interval $|\omega| < \delta_O$ and zero outside, where $\delta_O$ is a a constant that depends on the operator $\mathcal{O}$.[6]

If $\delta_O > \Delta E$, $\left| f(\bar{E}, \omega) \right|^2$ can be taken out of the sum and the result of (45) is simply $|f|^2$. Next we consider $T < \delta_O < \Delta E$. We approximate the sum with an integral

$$\langle \mathcal{O}_L \mathcal{O}_R \rangle_{\text{MCD}} \approx e^{-S(E)} \int_{E-\Delta E}^{E} \frac{dE_n}{T} \frac{dE_m}{T} e^{S(E_n)} e^{S(E_m)} \left| f(\bar{E}, \omega) \right|^2 e^{-S(\bar{E})}. \tag{46}$$

Here we had to pick a constant with dimension of energy to divide the infinitesimals $dE_n$ and $dE_m$: picking $T$ corresponds to have $d \approx e^{S(E)}$, if $\Delta E > T$. Next we change coordinates to $\bar{E} = (E_n + E_m)/2$ and $\omega = E_m - E_n$ and expand $S(E)$ around $\bar{E}$

$$\langle \mathcal{O}_L \mathcal{O}_R \rangle_{\text{MCD}} \approx e^{-S(E)} \int_{E-\Delta E}^{E} \frac{d\bar{E}}{T} e^{S(\bar{E})} \int_{-h(\bar{E})}^{h(\bar{E})} \frac{d\omega}{T} e^{S''(\bar{E})\frac{\omega^2}{4}} \left| f(\bar{E}, \omega) \right|^2. \tag{47}$$

The limits of integration for $\omega$ are given by

$$h(\bar{E}) = \begin{cases} 2(\bar{E} - E + \Delta E) & \bar{E} \leq E - \Delta E/2, \\ 2(E - \bar{E}) & \bar{E} > E - \Delta E/2. \end{cases} \tag{48}$$

---

[5]This difference was pointed out to us by D. Harlow [26], who also gave a version of the argument that follows.

[6]This can only be an approximation because the function $f(\bar{E}, \omega)$ should be smooth. To be more precise, we could take the function to exponentially decay for $\omega > \delta_O$, rather than being 0. This doesn't change the conclusion.

As long as $\Delta E^2$ is subextensive in the number of degrees of freedom - i.e. it scales like $S(E)^\alpha$ with an exponent $\alpha < 1$ - we can neglect the exponential in the second integral. The integral over $\omega$ can then be easily evaluated, we find

$$\langle \mathcal{O}_L \mathcal{O}_R \rangle_{\text{MCD}} \approx \frac{2}{T^2} \int_0^{\Delta E} d\varepsilon \, e^{-\beta \varepsilon} \min(\delta_O, h(E - \varepsilon)) \,. \tag{49}$$

Here we have also changed coordinates to $\varepsilon = E - \bar{E}$ and we have expanded $S(\bar{E})$ around $E$. The exponential in the last integral allows us to restrict the integration to $0 < \varepsilon < T$, where $h(E - \varepsilon) = 2\varepsilon$. As long as $\delta_O > T$ the result is again $|f|^2$, up to some numerical factor.

The intuitive reason why this result holds is that most states in the microcanonical window are within a distance $T$ from the top of the window. This follows from the fact that for small enough energy windows we can approximate the density of state with an exponential. More explicitly consider the energy variance, $\sigma_E^2 = \langle H^2 \rangle - \langle H \rangle^2$, in the microcanonical ensemble

$$\sigma_E^2 = \frac{1}{S'(E)^2} - \Delta E^2 \frac{e^{-S'(E)\Delta E}}{(1 - e^{-S'(E)\Delta E})^2} \,. \tag{50}$$

We see that for any $\Delta E \gg T$ the uncertainty is given by $T$ and not $\Delta E$ as one might naively think.

Let's now compare the result for the microcanonical double with the one for the thermofield double. Repeating the same steps as above we find

$$\langle \text{TFD} | \mathcal{O}_L \mathcal{O}_R | \text{TFD} \rangle = \frac{1}{Z(\beta)} \sum_{n,m} e^{-\beta \bar{E} - S(\bar{E})} \left| f(\bar{E}, \omega) \right|^2 \,. \tag{51}$$

As above, we now approximate the sum with an integral and change to $\bar{E}, \omega$ variables,

$$\langle \mathcal{O}_L \mathcal{O}_R \rangle_{\text{TFD}} \approx \frac{1}{Z(\beta)} \int \frac{d\bar{E}}{T^2} e^{-\beta \bar{E} + S(\bar{E})} \int_{-2\bar{E}}^{2\bar{E}} \left| f(\bar{E}, \omega) \right|^2 \approx \frac{\delta_O}{T} |f|^2 \,. \tag{52}$$

In the last step we have used that the first integral is dominated by energies $\bar{E} \gg \delta_O$, so we can restrict the second integral to $|\omega| < \delta_{\mathcal{O}}$ and take $f$ outside. The final answer now depends on $\delta_O$, so we conclude that, for the type of operators we are considering, the correlations in the thermofield double and microcanonical double are *not* the same. The class of operators we have chosen is general enough that the usual eternal black hole cannot be taken to be a good dual also for the microcanonical double.

This result seems at odds with the work of Marolf [27], in which he argued that the dual of a microcanonical version of the thermofield double is given by the usual eternal black hole. As we will explain in more detail below, the state considered in [27] is not the same as the microcanonical double state we have defined above, in the regime where semiclassical gravity applies. Specifically, while our state has equal probability for all energy eigenstates within the microcanonical window, in the state of [27] the probabilities are given by the Boltzmann factor. Further, the microcanonical window must be wide enough for the Boltzmann factor to become significant in order for the construction of [27] to be reliable.

In more detail, the state considered in [27], dubbed the "microcanonical thermofield double", is given by

$$| \text{MCTFD} \rangle = \sum_E e^{-\beta E/2} g(E - E_0) | E \rangle | E \rangle \,. \tag{53}$$

The function $g(E - E_0)$ enforces the microcanonical constraint; it is a smooth function that is approximately constant for $|E - E_0| < \Delta E$ and decays fast outside this range. For example [27]

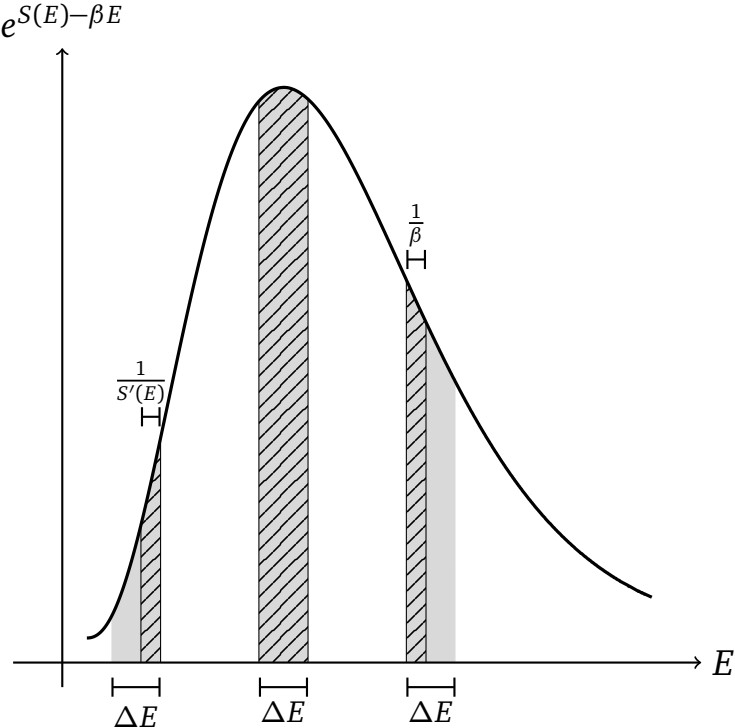

Figure 1: Statistical ensemble corresponding to $|\text{MCTFD}\rangle$. The function $g$ selects a window of energy states in an auxiliary Boltzmann distribution. We pick three windows (shaded areas), corresponding to the regimes explained in the text. From left to right: $\beta \ll S'(E_0)$, $(S'(E_0) - \beta)\Delta E \ll 1$ and $\beta \gg S'(E_0)$. The dashed areas correspond to the states that dominate the ensemble in the different regimes. The energy uncertainty is determined by these states only.

considers a Gaussian. However, the specific shape of the function is not important; in particular one can consider functions $g$ which approximate a step function. Then the only difference between our state and the one considered in [27] is the presence of the Boltzmann factor $e^{-\beta E/2}$.

Notice that in the state $|\text{MCTFD}\rangle$ $\beta$ does not correspond to the physical temperature of the system, which is instead given by $S'(E_0)$. To see this we can trace over one side of $|\text{MCTFD}\rangle$ to obtain the corresponding ensemble. Then it's clear that the function $g$ selects a window over an auxiliary Boltzmann distribution and $\beta$ is only a parameter of this distribution. Depending on how we choose $\beta$ and $E_0$ the resulting ensemble can be different. To see this we compute the energy variance in $|\text{MCTFD}\rangle$

$$\sigma_E^2 = \frac{1}{(S'(E_0) - \beta)^2} - \Delta E^2 \frac{e^{-(S'(E_0) - \beta)\Delta E}}{(1 - e^{-(S'(E_0) - \beta)\Delta E})^2} \,. \tag{54}$$

This is the same result we've found in (50) if we substitute $S'(E) \to S'(E_0) - \beta$. We recognize 3 different regimes for the resulting ensemble, which we illustrate in fig. 1. For $\beta \ll S'(E_0)$ we can neglect $\beta$ and we recover (50); if additionally $S'(E_0)\Delta E \gg 1$ the variance is given by the physical temperature squared $S'(E_0)^{-2}$. For $\beta \gg S'(E_0)$ we can neglect $S'(E_0)$ and we find that the variance is given by $\beta^{-2}$. Finally for $(S'(E_0) - \beta)\Delta E \ll 1$ the Boltzmann factor flattens the density of states and we find $\sigma_E^2 \propto \Delta E^2$.

The regime considered in [27] is the last one, $(S'(E_0) - \beta)\Delta E \ll 1$. Our state $|\text{MCD}\rangle$ instead corresponds to the $\beta \ll S'(E_0)$ regime, as there is no Boltzmann factor to begin with. Therefore, there is no inconsistency with [27].

One reason why the condition $(S'(E_0) - \beta)\Delta E \ll 1$ is necessary is the following. As explained in [27] to have a semiclassical dual geometry the uncertainty in the energy needs to be large enough. This follows from the energy-time uncertainty principle

$$\sigma_E \sigma_t > 1 \,. \tag{55}$$

To have a well defined dual geometry we want the uncertainty in time to vanish in the semiclassical limit $G_N \to 0$. This is true if the uncertainty in energy grows in this limit, $\sigma_E \propto G_N^{-\alpha}$, for some positive $\alpha$. In fact it should not be larger than in the canonical ensemble, so we also need $\alpha < 1/2$. For our state $|\text{MCD}\rangle$ the uncertainty is set by the temperature $\sigma_E = 1/S'(E_0)$, which doesn't scale with $G_N$. The uncertainty on time is larger than the thermal scale and the dual state is given by a superposition of distinct geometries. Of the three regimes above the only one that allows to tune the energy variance by changing $\Delta E$ is the last one, $(S'(E_0) - \beta)\Delta E \ll 1$, which is the one considered in [27].

The conclusion of the above discussion is that $|\text{MCD}\rangle$ does not have a semiclassical holographic dual. In the next section we introduce a larger class of averaging ensembles that lead to doubled states with a semiclassical dual.

# 4 More general ensembles

In the last section we have considered microcanonical averages of pure states. The quantum deviation is computed by the connected two-point function in a microcanonical version of the thermofield double. Unfortunately, as we have shown, this state has no semiclassical holographic dual.

In this section we consider more general averages over pure states and check whether the quantum deviation can still be computed as the connected two-point function in a thermofield-like state. We then look for examples in which this state does have a semiclassical holographic dual. In particular we expect that for thermal averages the quantum deviation is computed in the usual thermofield double state, which is dual to an eternal black hole.

We would like to study the quantum deviation, once we have fixed $\rho_1$ to be given by a specific state

$$\rho_1 = \sum_n \frac{\hat{p}_n}{Z_1} |n\rangle\langle n| \,. \tag{56}$$

Here $\hat{p}_n$ is an unnormalized probability distribution with normalization $Z_1 = \sum_n \hat{p}_n$ and $|n\rangle$ are energy eigenstates. For examples, in the canonical ensemble the first would be given by Boltzmann factors, $\hat{p}_n = e^{-\beta E_n}$, and the second by the partition function, $Z_1 = Z(\beta)$. The requirement (56) fixes the two-point function as

$$[\![\psi_n \psi_m^*]\!] = \frac{\hat{p}_n}{Z_1} \delta_{nm} \,. \tag{57}$$

Physically $\hat{p}_n/Z_1$ is the probability of finding the system in the state $n$. Notice that neither $\hat{p}_n$ nor $Z_1$ are physical: we can rescale $\hat{p}_n$ by a $n$-independent constant without changing the physical probabilities. The reason why we split them this way is that below we will deal with derived probability distributions with unnormalized probabilities $(\hat{p}_n)^k$ and normalization $Z_k = \sum_n (\hat{p}_n)^k$.

The constraint on the two-point function is not enough to uniquely select a measure over Hilbert space, since the higher point functions still need to be fixed. In fact there are an infinite number of possible probability distributions over pure states consistent with the constraint above. As an example, consider the microcanonical ensemble. This can be obtained from the

measure we have considered in the previous section, which is constant over the whole space spanned by the energy eigenstates in the window; or equivalently from a measure which is localized only on these energy eigenstates. We need some extra physics input to pick one. To compute the quantum deviation we only need the 4-point function, so it will be enough to determine this.

This problem is not new to the literature, see [23] and references therein. We comment on two alternative possible averaging procedures in app. B and C. To briefly summarize, we found the existing proposals lacking for our present purposes.

## 4.1 Nearly Gaussian ensembles

We now show that for a large class of averages the quantum deviation is indeed computed by a connected two-point function in a thermofield-like state. Rather than trying to define an explicit measure over Hilbert space, we consider directly the 4-point function needed to compute $\rho_2$ and determine it by imposing some physically motivated constraint. We come back to the actual integral over Hilbert space in the next section.

We assume that the probability distribution over states is such that, in the basis $|n\rangle$, the probability for the state $\sum_n \psi_n |n\rangle$ depends only on the magnitude of $\psi_n$ and not its phase. This was already true for the two-point function in (57), we are now assuming that this is true also for all higher point functions. Then the only non-zero terms are when the indices contract, and $\rho_2$ takes the simpler form

$$\rho_2 = \sum_{n,m} \frac{[\![|\psi_n|^2 |\psi_m|^2]\!]}{1+\delta_{nm}} \big( |n\,m\rangle\langle n\,m| + |n\,m\rangle\langle m\,n| \big). \tag{58}$$

In the ensembles of interest we expect that the probability distribution over the amplitudes will be close to Gaussian, so it is helpful to separate the Gaussian part. Define

$$[\![|\psi_n|^2 |\psi_m|^2]\!] \equiv [\![|\psi_n|^2]\!][\![|\psi_m|^2]\!](1+\delta_{nm}) + \varepsilon_{nm}, \tag{59}$$

where $\varepsilon_{nm}$ captures the deviation from the Gaussian theory. Then the Gaussian part simplifies, so that

$$\rho_2 = \sum_{n,m} \Big( [\![|\psi_n|^2]\!][\![|\psi_m|^2]\!] + \frac{\varepsilon_{nm}}{(1+\delta_{nm})} \Big) \big( |n\,m\rangle\langle n\,m| + |n\,m\rangle\langle m\,n| \big). \tag{60}$$

Notice that by definition we have $\mathrm{tr}_R \rho_2 = \rho_1$, which also guarantees that $\rho_2$ is properly normalized. This is true if $\varepsilon_{nm}$ satisfies

$$\sum_m \varepsilon_{nm} = -\Big( \frac{\hat{p}_n}{Z_1} \Big)^2. \tag{61}$$

This for example rules out a purely Gaussian ensemble, $\varepsilon_{nm} = 0$. Physically, we are defining a distribution over normalized states, and the normalization constraint couples the amplitudes $\psi_n$, leading to a nontrivial 4-point function.

The quantum deviation is given by

$$\Delta_O^2 = \sum_{n,m} \frac{\hat{p}_n \hat{p}_m}{Z_1^2} |\mathcal{O}_{nm}|^2 + \sum_{n,m} \frac{\varepsilon_{nm}}{(1+\delta_{nm})} \big( \mathcal{O}_{nn}^* \mathcal{O}_{mm} + |\mathcal{O}_{nm}|^2 \big). \tag{62}$$

To proceed further we need some way to fix $\varepsilon_{nm}$. To do this it is helpful to consider the non-Gaussian part for the microcanonical ensemble,

$$\varepsilon_{nm}^{mc} = -\frac{1}{d^2(d+1)}(1+\delta_{nm}). \tag{63}$$

The $\varepsilon_{nm}$ for the more general ensembles we are considering in this section should reduce to this expression when we specialize to microcanonical probabilities, $\hat{p}_n = 1$ and $Z_1 = d$. However, it would be too much to ask to find $\varepsilon_{nm}$ exactly, as we did for the microcanonical case. It is enough to determine it to leading order in $1/d$, which is the natural large parameter in the calculation. To leading order in this expansion the microcanonical non-Gaussian piece is given by

$$\varepsilon_{nm}^{mc} = -\frac{1}{d^3}(1 + \delta_{nm}) + O\left(\frac{1}{d^4}\right). \tag{64}$$

To decide how to replace $1/d^3$ in the more general ensembles we impose a physical constraint. Consider the quantum deviation of the Hamiltonian itself

$$\Delta_H^2 = \frac{1}{Z_1^2}\sum_n E_n^2 \hat{p}_n^2 + \sum_{nm} \varepsilon_{nm} E_n E_m. \tag{65}$$

This measures how much the energy expectation value varies across the ensemble we consider. In the microcanonical ensemble it is given by

$$\Delta_H^2 = \frac{1}{d}\left[\frac{1}{d}\sum_n E_n^2 - \left(\frac{1}{d}\sum_n E_n\right)^2\right] - \frac{1}{d^3}\sum_n E_n^2 + \cdots = \frac{1}{d}\sigma_E^2 + O\left(\frac{1}{d^2}\right). \tag{66}$$

We see that up to the prefactor and higher order corrections it is given by the energy variance in the microcanonical ensemble. We want this to be true also for the more general ensembles we consider in this section. This is enough to fix the 4-point function, i.e. $\varepsilon_{nm}$ to leading order.

Notice that because in the first term in eq. (65) we have $\hat{p}_n^2$, rather than simply $\hat{p}_n$, we are naturally led to consider the energy fluctuations in a different ensemble than the one we have introduced with $\rho_1$. We consider the ensemble with unnormalized probabilities $\hat{p}_n^2$ and normalization $Z_2 = \sum_n \hat{p}_n^2$. To make this clearer, let's specialize for a moment to canonical probabilities, $\hat{p}_n = \exp(-\beta E_n)$. Then the difference between the two ensembles is the temperature, which in the second one is half the original one, $\beta \to 2\beta$. Notice that the microcanonical ensemble is not sensitive to this difference, since in this case $\hat{p}_n^2 = \hat{p}_n$ and $Z_2 = Z_1$. Therefore, this choice is consistent with the microcanonical case. While our discussion below is valid for more general choices of $\hat{p}_n$, we will still sometimes refer to these two ensembles as being at temperature $\beta$ and $2\beta$.

The $\varepsilon_{nm}$ selected by the requirement above is

$$\varepsilon_{nm} = -\frac{Z_2}{Z_1^2}\frac{\hat{p}_n^2 \hat{p}_m^2}{Z_2^2}(1 + \delta_{nm}) + \varepsilon_{nm}^{(1)}, \tag{67}$$

where $\varepsilon_{nm}^{(1)}$ represents higher order corrections. It's easy to see that this expression reduces to (64) for $\hat{p}_n = 1$. We have written the expression above in such a way that each term is manifestly physical, by which we mean that it is invariant under $n$-independent rescaling of the unnormalized probabilities $\hat{p}_n$. Physically $Z_2/Z_1^2$ and $\hat{p}_n^2/Z_2$ are equal to respectively the purity and the normalized probabilities in the ensemble at temperature $2\beta$. We focus on probability distributions such that a large number, $e^S$, of states dominate the ensemble. In this case we can estimate both the purity and the physical probabilities as $O(e^{-S})$. We expect the higher order terms to be further suppressed by $O(e^{-S})$, i.e. we have $\varepsilon_{nm}^{(1)} = O(e^{-4S})$.

Now that we have fixed $\varepsilon_{nm}$ to leading order, we can go back and find the quantum deviation for a generic operator $\mathcal{O}$. From eq. (60) we have

$$\rho_2 = \rho_1 \otimes \rho_1 + \frac{Z_2}{Z_1^2}\left[\sum_{nm}\left(\frac{\hat{p}_n \hat{p}_m}{Z_2}|n\,m\rangle\langle m\,n| - \frac{\hat{p}_n^2 \hat{p}_m^2}{Z_2^2}|n\,m\rangle\langle n\,m|\right) - \sum_{nm}\frac{\hat{p}_n^2 \hat{p}_m^2}{Z_2^2}|n\,m\rangle\langle m\,n|\right] + \dots, \tag{68}$$

where the dots represent the corrections coming from $\varepsilon_{nm}^{(1)}$. We can rewrite this expression as

$$\rho_2 = \rho_1 \otimes \rho_1 + \frac{Z_2}{Z_1^2}\Big[\big((\Theta_L^\dagger |T_2\rangle\langle T_2| \Theta_L)^{T_L} - \tau_2 \otimes \tau_2\big) - \frac{Z_4}{Z_2^2}\big(\Theta_L^\dagger |T_4\rangle\langle T_4| \Theta_L\big)^{T_L}\Big] + \dots, \quad (69)$$

where $T_L$ denotes the partial transpose, and we have defined the thermofield-like state

$$|T_k\rangle \equiv \frac{1}{\sqrt{Z_k}} \sum_n \hat{p}_n^{k/2} |\tilde{n}\rangle |n\rangle, \qquad (70)$$

and the associated reduced density matrix

$$\tau_k \equiv \mathrm{Tr}_R |T_k\rangle\langle T_k| = \frac{1}{Z_k} \sum_n \hat{p}_n^k |n\rangle\langle n| . \qquad (71)$$

When computing the quantum deviation, the terms in the round parentheses in (69) lead to a connected two-point function in the state $|T_2\rangle$. We have

$$\Delta_O^2 = \frac{Z_2}{Z_1^2}\Big(\langle T_2| \mathcal{O}_L \mathcal{O}_R |T_2\rangle_c - \frac{Z_4}{Z_2^2} \langle T_4| \mathcal{O}_L \mathcal{O}_R |T_4\rangle + \dots\Big), \qquad (72)$$

where $\langle \cdot \rangle_c$ denotes the connected two-point function

$$\langle T_2| \mathcal{O}_L \mathcal{O}_R |T_2\rangle_c \equiv \langle T_2|(\mathcal{O}_L - \langle \mathcal{O}_L\rangle_2)(\mathcal{O}_R - \langle \mathcal{O}_R\rangle_2)|T_2\rangle, \qquad (73)$$

and $\langle \mathcal{O}\rangle_k = \mathrm{Tr}[\tau_k \mathcal{O}]$.

Typically $Z_4/Z_2^2 \sim e^{-S}$; since also $\varepsilon_{nm}^{(1)}$ is exponentially suppressed in entropy, we expect that the connected correlation function in $|T_2\rangle$ gives the dominant contribution to the quantum deviation, unless the operator $\mathcal{O}$ is quite unusual. We can write

$$\Delta_O^2 = \frac{Z_2}{Z_1^2}\Big(\langle T_2| \mathcal{O}_L \mathcal{O}_R |T_2\rangle_c + O(e^{-S})\Big), \qquad (74)$$

which is the generalization of (38) we were after.

Below we will discuss more precisely when the exponentially small corrections can become important. However, first we point out two specific ensembles where the state $|T_2\rangle$ has a semiclassical holographic dual.

**Example 1: canonical ensemble.** As a first example we consider $\hat{p}_n = e^{-\beta E_n}$. Then the state $|T_2\rangle$ is the thermofield double at temperature $2\beta$ and the quantum deviation is

$$\Delta_O^2 = \frac{Z(2\beta)}{Z(\beta)^2}\big[\langle \mathrm{TFD}_{2\beta}| \mathcal{O}_L \mathcal{O}_R |\mathrm{TFD}_{2\beta}\rangle_c + O(e^{-S})\big]. \qquad (75)$$

At sufficiently high temperatures the thermofield double is dual to the eternal black hole [28]. Thus, we conclude that the quantum deviation is given by the connected piece of a two-sided correlation function in an eternal black hole background.

**Example 2: mesocanonical ensemble.** Now that we have a canonical average, we can repeat the construction of Marolf [27] to obtain a microcanonical-like ensemble which does have a gravity dual. To distinguish this ensemble from the usual microcanonical ensemble, we call it *mesocanonical*. We consider probabilities

$$\hat{p}_n = e^{-\beta E_n/2} g(E_n - E_\beta), \qquad (76)$$

where the function $g$ selects a window of the energy distribution as in fig. 1. The window is centred at the energy $E_\beta$, which solves the saddle-point equation $S'(E) = \beta$, and has a width $\Delta E \gg T$. As we've explained in the previous section this is needed to have a semiclassical dual. Then by our discussion above we know that the quantum deviation is given by a connected two-point function in the state $|T_2\rangle$ which is given by

$$|T_2\rangle = \frac{1}{Z_2} \sum_n e^{-\beta E_n/2} g(E_n - E_\beta) |\tilde{n}\rangle |n\rangle . \tag{77}$$

This is precisely the state considered in [27], where it was shown that the dual geometry is the usual eternal black hole.

**Normalization from Gravity.** One might be annoyed by the presence of the explicit factor $Z_2/Z_1^2$ appearing in our expression (74) for the quantum deviation. In the above cases, this factor can be calculated in gravity separately. Perhaps more elegantly, we can choose to normalize states in the 'gravity normalization' [6]

$$\langle E|E\rangle = e^{S(E)} . \tag{78}$$

In this normalization, (74) becomes simply

$$(\Delta_O^2)_{\text{Grav}} = \langle T_2| \mathcal{O}_L \mathcal{O}_R |T_2\rangle_{\text{Grav}} + \dots , \tag{79}$$

where the subscript 'Grav' indicates that the states are now normalized using the gravity normalization.

## 4.2 Conditions for the validity of (74)

We present two conditions that together are sufficient to guarantee the validity of our formula (74). To do this it is convenient to split the quantum deviation in three pieces, $\Delta_O^2 = \text{I} - \text{II} + \text{III}$ where

$$
\begin{aligned}
\text{I} &= \frac{Z_2}{Z_1^2} \langle T_2| \mathcal{O}_L \mathcal{O}_R |T_2\rangle_c = \frac{1}{Z_1^2} \sum_{nm} \hat{p}_n \hat{p}_m |\mathcal{O}_{nm}|^2 - \frac{1}{Z_1^2 Z_2} \sum_{nm} \hat{p}_n^2 \hat{p}_m^2 (\mathcal{O}_{nn} \mathcal{O}_{mm}^*) , \\
\text{II} &= \frac{Z_4}{Z_1^2 Z_2} \langle T_4| \mathcal{O}_L \mathcal{O}_R |T_4\rangle = \frac{1}{Z_1^2 Z_2} \sum_{nm} \hat{p}_n^2 \hat{p}_m^2 |\mathcal{O}_{nm}|^2 , \\
\text{III} &= \sum_{nm} \tilde{\varepsilon}_{nm}^{(1)} \big( |\mathcal{O}_{nm}|^2 + \mathcal{O}_{nn} \mathcal{O}_{mm}^* \big) .
\end{aligned}
\tag{80}
$$

Notice that the first two terms are, by definition, greater or equal to zero, $\text{I}, \text{II} \geq 0$. The last term, III, comes from the higher order contributions to $\varepsilon_{nm}$. To lighten the notation we have defined $\tilde{\varepsilon}_{nm}^{(1)} \equiv (1 + \delta_{nm}) \varepsilon_{nm}^{(1)}$.

We should not trust our formula if $\text{I} < |\text{II} - \text{III}|$. We don't know the precise form of $\tilde{\varepsilon}_{nm}^{(1)}$, but we expect that it is exponentially suppressed in the entropy as compared to the leading non-gaussian piece, namely

$$\tilde{\varepsilon}_{nm}^{(1)} \lesssim e^{-S} \frac{1}{Z_1^2 Z_2} \hat{p}_n^2 \hat{p}_m^2 . \tag{81}$$

One might think that from this follows that III can always be neglected, but this is not the case. For example, whenever $\text{I} \leq \text{II}$, we also need to have $\text{III} \geq \text{II}$, because the quantum deviation is by definition greater or equal to zero. More generally, it is certainly true that $\sum_{nm} \tilde{\varepsilon}_{nm}^{(1)} |\mathcal{O}_{nm}|^2 \ll \text{II}$, and so we don't need to worry about the first term in III. However, we need to be more careful with the second term. Consider, for example, a diagonal operator,

$\mathcal{O}_{nm} \propto \delta_{nm}$. The $\delta_{nm}$ kills one sum in II, but it doesn't affect the second term in III. The extra sum in this term can compensate the exponential suppression in $\tilde{\varepsilon}^{(1)}_{nm}$.

To be explicit, consider a diagonal operator $\mathcal{O}$ with constant entries, $\mathcal{O}_{nm} = \lambda \delta_{nm}$. It's easy to see that $I = 0$ in this case; what is left in the quantum deviation is

$$\Delta_{\mathcal{O}}^2 = \lambda^2 \Big( -\frac{Z_4}{Z_1^2 Z_2} + \sum_{nm} \tilde{\varepsilon}^{(1)}_{nm}(1 + \delta_{nm}) \Big). \tag{82}$$

However, for a constant operator the quantum deviation should be zero, since the expectation value of the operator in any state is the same. This implies that the two terms above cancel out, which can be shown to be indeed the case using the constraint (61). In this example we explicitly see that III can be as important as II. In particular we can't exclude the possibility that for some operators, $|II - III| > I$ even if $II < I$. Notice that in this particular example the opposite is true: while $II > I$, we have $|II - III| \not> I$. In other words III works in our favor and we can trust our formula even though $II > I$. It would be interesting to understand whether this is true for a larger class of operators $\mathcal{O}$.

We sidestep this issue by separately imposing that $II < I$ and $|III| < I$. The first condition is simple to check, we need to compare the connected two-point function in $|T_2\rangle$ with the full two-point function in $|T_4\rangle$. For the second condition we need to estimate III. Using (81), we find that the the second term in III, which is the only one that can give problems when $I < II$, can be approximated by the disconnected piece of the two-point function in $|T_2\rangle$ times a factor exponentially small in the entropy. The two conditions are then given by

$$\begin{aligned}
II < I: \quad & \langle T_2 | \mathcal{O}_L \mathcal{O}_R | T_2 \rangle_c \gg e^{-S} \langle T_4 | \mathcal{O}_L \mathcal{O}_R | T_4 \rangle, \\
|III| < I: \quad & \langle T_2 | \mathcal{O}_L \mathcal{O}_R | T_2 \rangle_c \gg e^{-S} \langle \mathcal{O}_L \rangle_2 \langle \mathcal{O}_R \rangle_2.
\end{aligned} \tag{83}$$

These two conditions are sufficient to trust our formula for the quantum deviation, but are not necessary. The simple example we considered above violates them both and yet the answer we get from our formula is correct.

## 5 Maximum entropy

In the previous section we have fixed the 4-point function of $\psi_n$ by imposing some physically motivated constraint, inspired by the microcanonical ensemble. This is not completely satisfying for two reasons. First, the constraint, while reasonable, might seem rather ad hoc. Second, we haven't explicitly constructed an ensemble of states which leads to such 4-point function. In this section we address these two points. We show that at least one well-motivated choice of an ensemble of pure states reproduces the 4-point function (67).

The starting point is the same as in the previous section, suppose the 2-point function is fixed,

$$[\![\psi_n \psi_m^*]\!] = \frac{\hat{p}_n}{Z_1} \delta_{nm}. \tag{84}$$

**Question**: what is the most natural ensemble of pure states that reproduces this 2-point function? One natural answer is to choose the ensemble to maximize the entropy of the ensemble.

Note that we are not discussing a von Neumann entropy here, but a notion from probability theory. As an elementary example, consider a probability distribution $f(x)$ over one real variable. Define the entropy by

$$S = -\int dx\, f \log f, \tag{85}$$

subject to the constraint

$$\langle x^2 \rangle = \sigma^2 . \tag{86}$$

The solution is that $f$ is a Gaussian.

More generally, this procedure works as long as we have a preferred integration measure (in the above case, simply $dx$). An integration measure is needed because the entropy defined above depends on the measure. This can be thought of as an anomaly that arises when we try to take the continuum limit of the usual definition of entropy.

In our case, the natural metric on Hilbert space is given by the Fubini-Study metric, or equivalently the metric on complex projective space $\mathbb{CP}_d$. This space can be obtained from $\mathbb{C}_{d+1}$ by first restricting to the sphere $\|\psi\|^2 \equiv \sum_n |\psi_n|^2 = 1$, and then projecting out the overall phase. For us, it is more convenient *not* to project out the overall phase, and instead work with the full space of normalized wavefunctions.

In other words, while $\mathbb{CP}_d$ is obtained by the quotient of the $2d+1$-sphere by $U(1)$, we work simply on the sphere $S_{2d+1}$ with the associated metric. Our probability distribution will turn out to be invariant under the $U(1)$ action, so we can equivalently think of it as a probability distribution on $S_{2d+1}$ or $\mathbb{CP}_d$.

Making use of the usual metric on the sphere, we want to maximize

$$S = -\int \Big( \prod_{l=1}^{d} d\psi_l^* d\psi_l \Big) \delta(\|\psi\|^2 - 1) f \log f , \tag{87}$$

subject to the constraints

$$[\![ |\psi_n|^2 ]\!] = \frac{\hat{p}_n}{Z_1} . \tag{88}$$

We can solve for $f$ by performing a constrained minimization, varying the function

$$\tilde{S} = \int \Big( \prod_{l=1}^{d} d\psi_l^* d\psi_l \Big) \delta(\|\psi\|^2 - 1) \Big( -f \log f - \sum_m \alpha_m |\psi_m|^2 f - \beta f \Big), \tag{89}$$

where the $\alpha_n$ and $\beta$ are Lagrange multipliers.

The solution is

$$f = A \exp\Big( -\sum_m \alpha_m |\psi_m|^2 \Big), \tag{90}$$

which, as promised, is invariant under rotation by an overal phase $\psi_m \to e^{i\phi} \psi_m$.

The $\alpha_n$ need to be fixed by imposing the constraint on the two-point function. To do this it is convenient to define a 'generating function'

$$J(\alpha_a) \equiv \int \Big( \prod_{l=1}^{d} d\psi_l^* d\psi_l \Big) \delta(\|\psi\|^2 - 1) f , \tag{91}$$

which using the above form of $f$ becomes

$$J = A \int \Big( \prod_{l=1}^{d} d\psi_l^* d\psi_l \Big) \delta(\|\psi\|^2 - 1) e^{-\sum_m \alpha_m |\psi_m|^2} . \tag{92}$$

From this expression we learn that the $\alpha_n$ are defined up to a constant shift, $\alpha_n \to \alpha_n + c$, where $c$ is some $n$-independent constant. This freedom will be useful further below. We can calculate $J$ by rewriting the delta function,

$$J = \int \Big( \prod_{l=1}^{d} d\psi_l^* d\psi_l \Big) d\lambda \, e^{-\sum_m \alpha_m |\psi_m|^2} e^{-i\lambda(\sum_m |\psi_m|^2 - 1)} . \tag{93}$$

The two-point function is given by

$$\llbracket |\psi_n|^2 \rrbracket = \frac{1}{J} \int \Big( \prod_{l=1}^d d\psi_l^* d\psi_l \Big) d\lambda \, e^{i\lambda - \sum_m (\alpha_m + i\lambda)|\psi_m|^2} |\psi_n|^2 . \tag{94}$$

Integrate by parts on $\psi_n$ to obtain

$$\begin{aligned}
\llbracket |\psi_n|^2 \rrbracket &= \frac{1}{J} \int \Big( \prod_{l=1}^d d\psi_l^* d\psi_l \Big) d\lambda \, e^{i\lambda - \sum_m (\alpha_m + i\lambda)|\psi_m|^2} \frac{1}{\alpha_n + i\lambda} \\
&= \left\llbracket \frac{1}{\alpha_n + i\lambda} \right\rrbracket .
\end{aligned} \tag{95}$$

In this last equality, we are now thinking of $\lambda$ as one of the fields in the theory. We expand this expression in $\lambda$, and define $\beta_n \equiv \alpha_n^{-1}$, to get

$$\llbracket |\psi_n|^2 \rrbracket = \sum_{r \geq 0} (-i)^r \beta_n^{r+1} \llbracket \lambda^r \rrbracket = \frac{\hat{p}_n}{Z_1} . \tag{96}$$

From this we learn that to leading order $\beta_n = \hat{p}_n / Z_1$.

Similarly, for the 4-point function we have

$$\llbracket |\psi_n|^2 |\psi_m|^2 \rrbracket = (1 + \delta_{nm}) \left\llbracket \frac{1}{\alpha_n + i\lambda} \frac{1}{\alpha_m + i\lambda} \right\rrbracket . \tag{97}$$

Now we want to relate the 4-point function to the 2-point function. If we expand the expression above in $\lambda$ we find

$$\llbracket |\psi_n|^2 |\psi_m|^2 \rrbracket = (1 + \delta_{nm}) \sum_{r,s \geq 0} (-i)^{r+s} \beta_n^{r+1} \beta_m^{s+1} \llbracket \lambda^{r+s} \rrbracket . \tag{98}$$

We add and subtract the Gaussian piece, $(1 + \delta_{nm}) \hat{p}_n \hat{p}_m / Z_1^2$, from the expression above,

$$\llbracket |\psi_n|^2 |\psi_m|^2 \rrbracket = (1 + \delta_{nm}) \Big[ \frac{\hat{p}_n \hat{p}_m}{Z_1^2} + \sum_{r,s \geq 1} (-i)^{r+s} \beta_n^{r+1} \beta_m^{s+1} \big( \llbracket \lambda^{r+s} \rrbracket - \llbracket \lambda^r \rrbracket \llbracket \lambda^s \rrbracket \big) \Big] . \tag{99}$$

To leading order, we only need to keep the $r, s = 1$ term in the sum and we can set $\beta_n = \hat{p}_n / Z_1$,

$$\llbracket |\psi_n|^2 |\psi_m|^2 \rrbracket = (1 + \delta_{nm}) \Big[ \frac{\hat{p}_n \hat{p}_m}{Z_1^2} - \frac{\hat{p}_n^2 \hat{p}_m^2}{Z_1^4} \big( \llbracket \lambda^2 \rrbracket - \llbracket \lambda \rrbracket^2 \big) + \dots \Big] . \tag{100}$$

We now want to calculate $\llbracket \lambda^2 \rrbracket - \llbracket \lambda \rrbracket^2$. To do this, it is convenient to go back to the expression for the generating function (93) and perform the $\psi$ integrals, giving

$$J = \int d\lambda \, e^{i\lambda} \prod_l \frac{1}{\alpha_l + i\lambda} . \tag{101}$$

To proceed further we can rewrite the infinite product in terms of logarithms

$$J = \int d\lambda \, e^{i\lambda} \exp\Big( -\sum_l \log(1 + i\lambda \beta_l) \Big) . \tag{102}$$

Here we have dropped an irrelevant constant $\prod_l \beta_l$. As we have seen above, to leading order $\beta_l = \hat{p}_l / Z_1$. For typical probability distributions we expect $\hat{p}_l / Z_1 = O(1/d)$. It's clear that for

$\lambda\beta_l > 1$ the integrand in $J$ decays fast, so we expand the integrand as if $\lambda\beta_l < 1$. For the log we have

$$\sum_l \log(1 + i\lambda\beta_l) = \sum_k \frac{(i\lambda)^k}{k}(-1)^{k+1}B_k\,, \tag{103}$$

where we have defined $B_k \equiv \sum_l \beta_l^k$. Notice that $B_k = O(d^{1-k})$, so to leading order we can focus on the linear and quadratic terms,

$$J = \int d\lambda\, e^{-\frac{1}{2}\lambda^2 B_2 + i\lambda(1-B_1)} + \dots\,. \tag{104}$$

To leading order we have $B_2 = Z_2/Z_1^2$ and $B_1 = 1$; the latter implies that the linear term drops out. However, it is not clear whether keeping higher order terms in $B_1$ gives a correction negligible as compared to the quadratic term. We can avoid this question by using the freedom in $\alpha_n$ we pointed out at the beginning of the section: we pick $c$ in such a way that $B_1 = 1$ to all orders. Finally we have that

$$J = \int d\lambda\, e^{-\frac{1}{2}\lambda^2 Z_2/Z_1^2} + \dots\,, \tag{105}$$

from which follows that $[\![\lambda^2]\!] - [\![\lambda]\!]^2 = Z_1^2/Z_2$.

Plugging this back in gives the final answer,

$$[\![|\psi_n|^2|\psi_m|^2]\!] = (1 + \delta_{nm})\left[\frac{\hat{p}_n\hat{p}_m}{Z_1^2} - \frac{Z_2}{Z_1^2}\frac{\hat{p}_n^2\hat{p}_m^2}{Z_2^2} + \dots\right]\,, \tag{106}$$

which agrees with (67).

# 6 Example: $\mathcal{O} = A(t)A(0)$

We consider the quantum deviation of $A(t)A(0)$. This operator is interesting because it can be used to study information loss in the bulk [28, 29].

Consider, in a holographic theory, a state $|\psi\rangle$ dual to a black hole geometry. The two point function can be computed by turning on sources at the boundary of the black hole geometry: generically it decays exponentially in time [30]. In a unitary theory with a finite dimensional Hilbert space the decay cannot continue forever. At times of order the entropy of the system the two-point function starts oscillating erratically around a value exponentially small in the entropy. This late time dynamics is invisible in semiclassical gravity [31, 32], which instead predicts an eternal exponential decay.

The goal of this section is to show that we can reliably compute the quantum deviation of $A(t)A(0)$ in semiclassical gravity, even for times $t \sim S$. Knowing the quantum deviation gives us access to information on the late time behavior of $A(t)A(0)$ which are otherwise invisible in semiclassical gravity.

Following [1] we assume that the operator $A$ is simple, meaning that it is insensitive to the details of the state $|\psi\rangle$. For these operators it is sensible to approximate the expectation value in $|\psi\rangle$ with its average over an ensemble of pure states drawn from a small energy window. We can achieve this with the mesocanonical ensemble as in (76),

$$\hat{p}_n = e^{-\beta E_n/2}g(E_n - E_\beta)\,. \tag{107}$$

Here $\beta$ is picked such that $S'(E_\beta) = \beta$ and $g$ is some function which selects the energy window. We take it to be a step function

$$g(E - E_\beta) = \begin{cases} 1 & E_\beta - \Delta E \leq E \leq E_\beta \,, \\ 0 & \text{otherwise} \,. \end{cases} \tag{108}$$

The width of the window is $\Delta E \gg T$, such that the geometries dual to the state $|T_2\rangle$ is semi-classical. It is given by the usual two-sided black hole.

The quantum deviation tells us how far we can trust this approximation. As we show below, at late times the average expectation value is smaller than the quantum deviation and hence should not be trusted. In this regime the quantum deviation gives us an estimate of the typical size of the two-point function. Before turning to the semiclassical computation of the quantum deviation we estimate the late time behaviour of $A(t)A(0)$ in quantum mechanics. To do this we assume that the operator $A(0)$ obeys ETH, see (22). For simplicity we assume that $A(E) = 0$, equivalently we could consider connected two-point functions. The matrix elements of $\mathcal{O} = A(t)A(0)$ are then given by

$$\begin{aligned} \mathcal{O}_{nm} &= \sum_k e^{i(E_n - E_k)t} A_{nk} A_{km} \\ &= \sum_k e^{i(E_n - E_k)t} e^{-S(\bar{E}')/2} e^{-S(\bar{E}'')/2} f(\bar{E}', \omega') f(\bar{E}'', \omega'') R_{nk} R_{km} \,, \end{aligned} \tag{109}$$

where $\bar{E}' = (E_k + E_n)/2$, $\bar{E}'' = (E_k + E_m)/2$ and $\omega' = E_k - E_n$, $\omega'' = E_m - E_k$. Notice that the $k$ sum goes over all the energies.

We want to estimate the size of these matrix elements at late times. The time evolution of the matrix elements depends on the detailed shape of the function $f$, which is not fixed by ETH [17]. We can find a lower bound by focussing on very late times, $t > e^S$, when we can approximate all the oscillating phases as random variables. From gravity we know that the two-point function decays exponentially fast; as we show below from this follows that the lower bound is reached already at $t \sim S$.

For simplicity we focus on operator with a spectral width of size $T$. Namely, we assume that the function $f$ is approximately constant for $|\omega| < T$ and that it decays quickly to zero outside. The regions $|\omega'| < T$ and $|\omega''| < T$ have a significant overlap only if $|E_n - E_m| < T$. If we focus on this region, we can set $\bar{E}' = \bar{E}'' = \bar{E} \equiv (E_n + E_m)/2$ and $\omega' = \omega'' = \omega \equiv \omega_m - \omega_n$ in the arguments of $S$ and $f$. Then we can take $e^S$ and $f$ out of the sums. We are left with

$$\mathcal{O}_{nm} \approx e^{-S(\bar{E})} f(\bar{E}, \omega)^2 \sum_k e^{i(E_n - E_k)t} R_{nk} R_{km} \,, \tag{110}$$

where now the $k$ sum goes only over energies within a distance $T$ from both $E_n$ and $E_m$. Notice that in this interval there are $e^{S(\bar{E})}$ states. To estimate this last sum we need to consider separately the diagonal and off-diagonal matrix elements of $\mathcal{O}$.

For the off-diagonal terms, $n \neq m$, the random numbers $R_{nk} R_{km}$ induce cancellations at all times. Since we are summing $e^{S(\bar{E})}$ terms, we can estimate the sum as $e^{S(\bar{E})/2}$.

For the diagonal terms, $n = m$, the random variables combine into a positive quantity $R_{nk} R_{kn} = |R_{nk}|^2$ and the time dependent phases play a more important role. The advantage of considering times $t > e^S$ is that we can approximate all the phases as random variables. We can again estimate the sum as $e^{S(\bar{E})/2}$.

We conclude that all the matrix elements can be approximated as

$$\mathcal{O}_{nm} \approx e^{-S(\bar{E})/2} f(\bar{E}, \omega)^2 \rho_{nm}(t) \,. \tag{111}$$

Here $\rho_{nm}(t)$ is a random complex variable erratically fluctuating in time around 0 with fluctuations that are, for most times, of order 1. For specific choice of $t$ Poincaré recurrences might occur and $\rho_{nm}(t)$ is much larger. To find this estimate we have assumed that $t > e^S$, so strictly speaking this is only a lower bound on the size of the matrix elements. However, from gravity we know that the two-point function decays exponentially in time. From this follows that this lower bound is already a good estimate at $t \sim S$.

Using the matrix elements above we can estimate both the two-point function in $|\psi\rangle$ and the averaged two-point function at late times. To estimate the first, let $|\psi\rangle = \sum_n \psi_n |n\rangle$, where $\psi_n$ are random complex numbers drawn according to the mesocanonical ensemble. The two point function is

$$\langle \psi | A(t)A(0) | \psi \rangle = \sum_{nm} \psi_n \psi_m^* e^{-S(E_n)/2} f(\bar{E}, \omega)^2 \rho_{nm}(t). \tag{112}$$

For typical states the effective number of terms contributing to each sum is given by $e^{S(E_\beta)}$. To see this notice that $e^{S(E_n)}\hat{p}_n$ is picked at $E_n = E_\beta$ and decays exponentially for $E_n < E_\beta - T$. In this interval there are $e^{S(E_\beta)}$ states. Therefore, we can approximate $\psi_n$ as a random vector where $e^{S(E_\beta)}$ components are of order $e^{-S(E_\beta)/2}$ and the remaining are negligible. Since the terms in sum don't have a definite sign there are cancellations and we can approximate the two sums with $e^{S(E_\beta)}$. Putting everything together we have

$$\langle \psi | A(t)A(0) | \psi \rangle \propto e^{-S(E_\beta)/2}. \tag{113}$$

The averaged two-point function is

$$[\![\langle \psi | A(t)A(0) | \psi \rangle]\!] = \frac{1}{Z_1} \sum_n e^{-\beta E_n/2} g(E_n - E_\beta) e^{-S(E_n)/2} f(E_n, 0)^2 \rho_{nn}(t). \tag{114}$$

The effective number of terms in the sum is again given by $e^{S(E_\beta)}$, so the averaged two-point function can be estimated as

$$[\![\langle \psi | A(t)A(0) | \psi \rangle]\!] \propto e^{-S(E_\beta)}. \tag{115}$$

We see that the averaged two-point function does not give a good approximation at late times.

Next we consider the quantum deviation. From eq. (62) we find

$$\Delta_O^2 = \frac{1}{Z_1^2} \sum_{n,m} \hat{p}_n \hat{p}_m |\mathcal{O}_{nm}|^2 - \frac{1}{Z_1^2 Z_2} \sum_{n,m} \hat{p}_n^2 \hat{p}_m^2 \left( \mathcal{O}_{nn}^* \mathcal{O}_{mm} + |\mathcal{O}_{nm}|^2 \right). \tag{116}$$

For the matrix elements above this quantity is dominated by the first term, so we focus on this. We approximate the sum over energies with an integral

$$\begin{aligned}
\Delta_O^2 &\approx \frac{1}{Z_1^2} \sum_{n,m} e^{-\beta(E_n+E_m)/2} g(E_n - E_\beta) g(E_m - E_\beta) |\mathcal{O}_{nm}|^2 \\
&\approx \frac{1}{Z_1^2} \int_{E_\beta - \Delta E}^{E_\beta} \frac{dE_n}{T} e^{S(E_n) - \beta E_n/2} \int_{E_\beta - \Delta E}^{E_\beta} \frac{dE_m}{T} e^{S(E_m) - \beta E_m/2} f(\bar{E}, \omega)^4 e^{-S(\bar{E})},
\end{aligned} \tag{117}$$

where in going to the second line we have neglected $|\rho_{nm}(t)|^2$. Next we change coordinates from $E_n, E_m$ to $\bar{E}, \omega$ and expanded $S(E_n), S(E_m)$ to first order around $\bar{E}$.

$$\begin{aligned}
\Delta_O^2 &\approx \frac{1}{Z_1^2} \int_{E_\beta - \Delta E}^{E_\beta} \frac{d\bar{E}}{T} e^{S(\bar{E}) - \beta \bar{E}} \int_{-h(\bar{E})}^{h(\bar{E})} \frac{d\omega}{T} f(\bar{E}, \omega)^4 \\
&\approx \frac{Z_2}{Z_1^2} f^4,
\end{aligned} \tag{118}$$

where $h(\bar{E})$ is the same as in eq. (48). In going to the last line we have evaluated the first integral by saddle point and used that $f$ decays fast for $|\omega| > T$. With similar calculations one can check that the remaining two terms in (116) are exponentially smaller.

The last thing we are left to do is estimating the prefactor. We calculate the $Z$'s by again approximating the sum over energy with an integral. For $Z_1$ we have

$$Z_1 \approx \int_{E_\beta - \Delta E}^{E_\beta} \frac{dE}{T} e^{S(E) - \beta E/2} \approx e^{S(E_\beta) - \beta E_\beta/2}, \tag{119}$$

and for $Z_2$

$$Z_2 \approx \int_{E_\beta - \Delta E}^{E_\beta} \frac{dE}{T} e^{S(E) - \beta E} \approx e^{S(E_\beta) - \beta E_\beta}. \tag{120}$$

Notice that the solution of the saddle point equation for the $Z_1$ lies outside the range of the integration so we have approximated the integral with its upper limit. This is possible because the integrand grows exponentially. We conclude that

$$\frac{Z_2}{Z_1^2} \propto e^{-S(E_\beta)}. \tag{121}$$

From this we can estimate the typical size of $\langle \psi | A(t) A(0) | \psi \rangle$ as $e^{-S(E_\beta)/2}$, which agrees with our previous calculation.

We now turn to the gravity dual. The state $\rho_1$ corresponds to the usual single-sided black hole. As we've said the two-point function exponentially decays in this geometry. From our analysis above we know that this can't be correct at sufficiently late times. In fact even if gravity could reliably compute the averaged two-point function at late times, this would not give a good estimate of $\langle \psi | A(t) A(0) | \psi \rangle$. This can be diagnosed by computing the quantum deviation which at late times is larger than the averaged two-point function. Interestingly we can reliably calculate $\Delta^2_{A(t)A(0)}$ in semiclassical gravity. The quantum deviation is given by

$$\Delta^2_{A(t)A(0)} = \frac{Z_2}{Z_1^2} \langle \mathrm{T}_2 | A_L(-t) A_L(0) A_R(t) A_R(0) | \mathrm{T}_2 \rangle_c. \tag{122}$$

We first consider the connected 4-point function and turn to the prefactor at the end.

For simplicity we specialize to $d = 2$ on the boundary. If we focus on operators with spectral width smaller than $\Delta E$ we can compute the correlation function by inserting the operators in the two-sided BTZ black hole, at temperature $\beta$. For this geometry the left-right propagator is known explicitly [33,34]:

$$\langle \mathrm{TFD}_\beta | A_L(t_1, \phi_1) A_R(t_2, \phi_2) | \mathrm{TFD}_\beta \rangle = \frac{(2\pi^2)^\Delta}{2\pi} \Big( \frac{\ell}{\beta} \Big)^{2\Delta} \tag{123}$$

$$\times \sum_{n \in \mathbb{Z}} \Big[ \cosh\Big( 2\pi \frac{\ell}{\beta} (\Delta\phi + 2\pi n) \Big) + \cosh\Big( 2\pi \frac{t_1 + t_2}{\beta} \Big) \Big]^{-\Delta}.$$

Here $\Delta$ is the scaling dimension of the operator $A$ and the sum over images is necessary to have the correct periodicity around the spatial circle. We will only consider operator insertions at the same point around the circle, $\Delta\phi = 0$, so from now on we suppress the dependence on $\phi$. We are using the convention in which time flows upwards on both sides of the geometry.

We can simplify the expression for the propagator if we focus on large black holes, $\beta \ll \ell$, for which the contribution from images is exponentially suppressed. We find

$$\langle \mathrm{TFD}_\beta | A_L(t_1) A_R(t_2) | \mathrm{TFD}_\beta \rangle \approx \Big( \frac{\ell}{\beta} \Big)^{2\Delta} \exp\Big( -2\pi\Delta \frac{|t_1 + t_2|}{\beta} \Big), \tag{124}$$

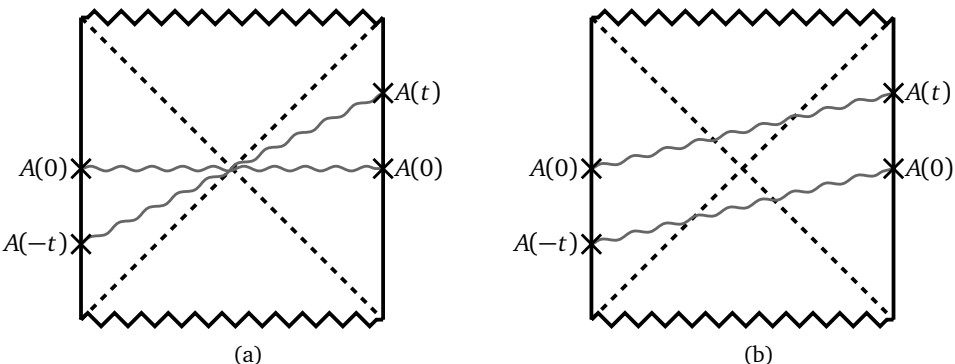

Figure 2: The two contractions that contribute to the connected correlation function. The contraction (a) is time independent, the contraction (b) decays exponentially in time.

where we have dropped the dimensionless prefactor. This approximation is good for $|t_1 + t_2| > \beta$.

To leading order in $G_N$ we can compute the 4-point function in (122) by contracting the fields as if they were freely propagating. Sometimes the relative boosts between the operators can lead to backreaction on the geometry, but for the time ordering we are considering this doesn't happen [35]. The contractions contributing to the connected correlation function are shown in fig. 2. Adding them we find

$$\left\langle \text{TFD}_\beta \big| A_L(-t)A_L(0)A_R(t)A_R(0) \big| \text{TFD}_\beta \right\rangle_c \approx \left(\frac{\ell}{\beta}\right)^{4\Delta}\left[1 + \exp\left(-4\pi\Delta\frac{t}{\beta}\right)\right]. \tag{125}$$

The exponentially decaying term is negligible for $t > \beta$, so we find that the correlation function is approximately given by a time-independent constant. This constant is of order $O(1)$, so we can neglect higher order corrections in $G_N$. The quantum deviation is then given by

$$\Delta^2_{A(t)A(0)} \approx \frac{Z_2}{Z_1^2}\left(\frac{\ell}{\beta}\right)^{4\Delta}. \tag{126}$$

This result matches with the quantum mechanical estimate of sec. 6 if we identify the constant $|f|$ with $(\ell/\beta)^\Delta$.[7]

From this we learn two things. First, the semiclassical expectation value of $A(t)A(0)$ should be trusted only until times $t \sim S$, after which the quantum deviation becomes larger. Second, the typical size of this expectation value, after this time, is given by $e^{-S/2}$. This we already knew from the quantum mechanical analysis above. The interesting point here is that by computing the quantum deviation this information is accessible also in semiclassical gravity.[8]

**Canonical ensemble.** One might wonder why above we haven't considered the canonical ensemble. In fact the calculation for the quantum deviation is almost the same, only the prefactor changes,

$$\Delta^2_O \propto \frac{Z(2\beta)}{Z(\beta)^2}. \tag{127}$$

---

[7] In the gravitational computation we also found an exponentially decaying piece. We believe that it is not possible to obtain this term from the standard version of ETH, where only the 2-point functions of $R_{nm}$ are not trivial. One needs to use the generalization to higher point functions of [36], see also [37] for a related work. However, we haven't looked at this carefully because in our case this extra term is uninteresting.

[8] In fact a similar statement holds also in the CFT. In the CFT the semiclassical limit corresponds to the infinite volume limit. In this limit, same as in gravity, the two-point function is given by an eternally decaying exponential. The quantum deviation, even in the infinite volume limit, gives us some information on the time at which this decay should stop and on the typical size of $A(t)A(0)$ at late times.

For the BTZ example we can estimate this ratio of partition functions as $e^{-3S/4}$. This is much larger than what we have found in the mesocanonical ensemble and it doesn't give a good estimate of the late-time fluctuations of $\langle\psi|A(t)A(0)|\psi\rangle$, for a fixed $|\psi\rangle$. Indeed in the canonical ensemble we are averaging over states with very different energies, and so the quantum deviation ends up being much larger. Notice that the same happens in standard statistical mechanics: the expectation values in the microcanonical and canonical ensembles are the same, but the fluctuations are different. If we want to say something about typical states at some energy $E_\beta$, we need to consider a sufficiently small energy window as we did above with the mesocanonical ensemble.

## 7  Discussion

In this paper, we have studied the *quantum deviation*

$$\Delta_O^2 \equiv \left[\!\left[\left|\operatorname{Tr}(\mathcal{O}\rho)\right|^2\right]\!\right] - \left|\left[\!\left[\operatorname{Tr}(\mathcal{O}\rho)\right]\!\right]\right|^2, \tag{128}$$

where the square double brackets indicate that we average over an ensemble of states and $\rho$ is a random state from the ensemble. This quantity is not the usual quantum variance and it gives us the fluctuations of the expectation value of the operator $\mathcal{O}$ in this ensemble. It is a diagnostic about whether or not we should trust the average of an expectation value, i.e. if the value of the quantum deviation is close to that of the average of the observable we should stop trusting the latter.

We found that the quantum deviation can be put in the form of a connected two-point function in a thermofield double-like state, for a large class of ensembles. So, in the context of AdS/CFT, the quantum deviation can be computed in semiclassical gravity whenever this state has a holographic dual. We have given two examples in which this is the case: the canonical and mesocanonical ensembles. For both these ensembles the gravity dual is the eternal AdS black hole. Specifically, in the canonical case the quantum deviation of an operator $\mathcal{O}$ is given by

$$\Delta_O^2 = \frac{Z(2\beta)}{Z(\beta)^2}\Big[\big\langle \text{TFD}_{2\beta}\big|(\mathcal{O}_L - \langle\mathcal{O}_L\rangle)(\mathcal{O}_R - \langle\mathcal{O}_R\rangle)\big|\text{TFD}_{2\beta}\big\rangle + O(e^{-S})\Big]. \tag{129}$$

As an example that illustrates how the quantum deviation can be useful, we calculated the quantum deviation of the operator $A(t)A(0)$ in the mesocanonical ensemble. At times $t \sim S$ the quantum deviation becomes larger than the averaged expectation value, which tells us that we should not trust the latter anymore. Moreover, the quantum deviation gives us an estimate of the typical fluctuations of $[\![\langle\psi|A(t)A(0)|\psi\rangle]\!]$ at late times, which are otherwise outside the reach of semiclassical gravity.

There are a number of directions one can take from here.

- **Purity.** We have argued, and checked in a simple example, that the quantum deviation "knows" about the purity of the ensemble of states we are using. It would be useful to examine if that is true more generally.

- **More general ensembles.** We have identified a class of ensembles of pure states where the quantum deviation can be calculated in gravity. It would be interesting to explore other ensembles of pure and mixed states. We suspect that the gravity calculation is reliable for a wider class of ensembles.[9]

---

[9] We understand that I. Arav, S. Chapman, and J. de Boer are examining this question in ongoing work [38].

- **Range of Validity.** The quantum deviation gives a diagnostic of whether the semiclassical calculation of the *mean* is reliable. Is there a diagnostic within semiclassical gravity that determines when the calculation of the quantum deviation is reliable?

- **Class of Observables.** We have provided two conditions, (83), to check whether our formula for the quantum deviation, (74), can be trusted, given a specific operator $\mathcal{O}$. These conditions are not very stringent, so we believe that our formula has a wide range of applicability. However, it would be interesting to check this more precisely. What are the precise requirements on the operator $\mathcal{O}$ such that the semiclassical calculation of the quantum deviation is reliable?

- **Fuzzballs.** It seems that our techniques could be used to place powerful constraints on fuzzball-type proposals. Namely, if the metric deviates from Schwarzschild outside the horizon in a typical pure state, this is constrained by the two point function in the eternal black hole background. It would be interesting to map out the constraints and apply them to various proposals.

- **Beyond AdS/CFT.** We would ultimately like to determine whether quantum gravity effects are observable in real black holes. To accomplish this, our results will need to be extended beyond the AdS/CFT context.

- **Polynomial tails.** The two-point functions of generic operators decay polynomially in time rather than exponentially, due to overlaps with conserved currents.[10] These tails overcome the exponential decay before the time scale $t \sim S$. In section 6, we didn't have to worry about these tails, because we have considered primaries in a 2d CFT. For these operators, there are no polynomial tails, and the two-point function decays exponentially until $t \sim S$. In general, we believe that to study information loss, it is more convenient to consider operators which don't have late-time polynomial tails. However, we haven't thought about which operators one should consider to avoid polynomial tails in higher dimensions. It would be interesting to address this question in future work.

## Acknowledgements

We gratefully acknowledge discussions with Igal Arav, Alex Belin, Jan de Boer, Shira Chapman, Oleksandr Gamayun, Beatrix Mühlmann, Yasunori Nomura, and especially James Sully. BF, DN, and AR are supported by the ERC Consolidator Grant QUANTIVIOL. This work is part of the $\Delta$ ITP consortium, a program of the NWO that is funded by the Dutch Ministry of Education, Culture and Science (OCW).

## A   Microcanonical averages

In this appendix we consider in more detail the microcanonical average over pure states we have used in sec. 3.

The ensemble is given by pure states in $\mathcal{H}_{\varepsilon} = \text{span}\{|n\rangle, E_n \in I\}$, where $|n\rangle$ are energy eigenstates and the microcanonical window is given by $I = [E - \Delta E, E]$. The states are drawn with equal probability. In practice to perform the average we integrate over the components

---

[10]We thank the anonymous referee for pointing this out to us.

of $\psi$ in the energy eigenbasis $\psi_n = \langle n|\psi\rangle$.[11] The integral is given by

$$\llbracket \dots \rrbracket \equiv \int_{\mathcal{H}_{\mathcal{E}}} \mathcal{D}\psi \,\dots, \tag{130}$$

with measure

$$\mathcal{D}\psi \equiv \frac{(d-1)!}{2\pi^d} \delta\big(1 - \|\psi\|\big) \prod_{l=1}^{d} \frac{d\psi_l d\psi_l^*}{2i}. \tag{131}$$

The constraint is needed to ensure that we only integrate over physical states, $\|\psi\|^2 = \langle\psi|\psi\rangle = 1$. Below we prove the following equation for the correlation functions of $\psi_n$,

$$\llbracket \psi_{n_1} \dots \psi_{n_k} \psi_{m_1}^* \dots \psi_{m_{k'}}^* \rrbracket = \delta_{k,k'} \frac{(d-1)!}{(d+k-1)!} \sum_{\sigma \in S_k} \delta_{n_1 m_{\sigma(1)}} \dots \delta_{n_k m_{\sigma(k)}}. \tag{132}$$

Here $\sigma$ is an element of the permutation group of $k$ elements, in other words we can simply use Wick contractions between the indices. In practice in the main text we only need the 2-point and 4-point functions, see sec. 3.

**Proof of eq. (130)**  We want to integrate over a complex hypersphere, the measure is given by

$$\llbracket \dots \rrbracket = \frac{1}{N} \int \prod_{l=1}^{d} \frac{d\psi_l^* d\psi_l}{2i} \delta\big(1 - \|\psi\|\big) \dots. \tag{133}$$

As a warm up let's compute the normalization

$$\begin{aligned}
\llbracket 1 \rrbracket &= \frac{1}{N} \int \prod_{l=1}^{d} \frac{d\psi_l^* d\psi_l}{2i} \delta\big(1 - \|\psi\|\big) \\
&= \frac{2}{N} \int \frac{dk}{\pi} e^{ik} \prod_{l=1}^{d} \int \frac{d\psi_l^* d\psi_l}{2i} e^{-ik\psi_l^*\psi_l} \\
&= \frac{2(-i\pi)^d}{N} \int \frac{dk}{\pi} \frac{e^{ik}}{k^d} = \frac{1}{N} \frac{2\pi^d}{(d-1)!},
\end{aligned} \tag{134}$$

from which we find $N = 2\pi^d/(d-1)!$, that is the volume of the unit $2d-1$ dimensional sphere. Above we have used $\delta(1 - \|\psi\|) = 2\delta(1 - \langle\psi|\psi\rangle)$ and Fourier transformed in going to the second line; we have used $\int d\psi^* d\psi \exp(-i\psi^*\psi) = 2\pi$ in going to the last line; finally we have used $\int_{\mathbb{R}} \exp(ik)k^{-n} = i^n\pi/(n-1)!$ in the last step.

To compute correlation functions we find the generating function

$$Z(J_l, J_l^*) \equiv \llbracket \exp\big(J_l\psi_l + J_l^*\psi_l^*\big) \rrbracket, \tag{135}$$

from which we can compute correlation functions taking $J$ derivatives. A computation similar to the one above shows that

$$Z(J_l, J_l^*) = (d-1)! \int \frac{dk}{\pi} \frac{e^{ik}}{(ik)^d} \exp\left(\frac{J_l^* J_l}{ik}\right). \tag{136}$$

---

[11]Alternatively, one could set $|\psi\rangle = U|m\rangle$, where $U$ is a Haar random unitary and $|m\rangle$ a reference state, e.g. the vacuum. The average over Hilbert space is then traded for one over the unitary group, which can be computed using Weingarten functions, as in [1]. We prefer averages over states because they are easier to work with.

As a simple example let's compute the two point function

$$
\begin{aligned}
[\![\psi_\alpha \psi_\beta^*]\!] &= \frac{\partial}{\partial J_\alpha}\frac{\partial}{\partial J_\beta^*}Z(J_l, J_l^*)\Big|_{J_l, J_l^*=0}\\
&= \delta_{\alpha\beta}(d-1)!\int \frac{dk}{\pi}\frac{e^{ik}}{(ik)^{d+1}} = \frac{1}{d}\delta_{\alpha\beta}\,.
\end{aligned}
\tag{137}
$$

The generalization to higher correlation functions is given by

$$
\begin{aligned}
[\![\psi_{\alpha_1}\dots\psi_{\alpha_k}\psi_{\beta_1}^*\dots\psi_{\beta_{k'}}^*]\!] &= \frac{\partial}{\partial J_{\alpha_1}}\dots\frac{\partial}{\partial J_{\alpha_k}}\frac{\partial}{\partial J_{\beta_1}^*}\dots\frac{\partial}{\partial J_{\beta_{k'}}^*}Z(J_l, J_l^*)\Big|_{J_l, J_l^*=0}\\
&= \frac{\partial}{\partial J_{\alpha_1}}\dots\frac{\partial}{\partial J_{\alpha_k}}J_{\beta_1}\dots J_{\beta_{k'}}(d-1)!\int\frac{dk}{\pi}\frac{e^{ik}}{(ik)^{d+k'}}\exp\left(\frac{J_l^* J_l}{ik}\right)\Big|_{J_l, J_l^*=0}\\
&= \delta_{kk'}\sum_{\sigma\in S_k}\delta_{\alpha_1\beta_{\sigma(1)}}\dots\delta_{\alpha_k\beta_{\sigma(k)}}\frac{(d-1)!}{(d+k-1)!}\,.
\end{aligned}
\tag{138}
$$

## B  Gaussian adjusted projected (GAP) measure

As we mentioned in sec. 4 the problem of finding a probability distribution over pure states that reproduces

$$
[\![\psi_m^*\psi_n]\!] = \frac{\hat{p}_n}{Z_1}\delta_{nm}
\tag{139}
$$

has been already studied in the literature. In this appendix we would like to compare our result with the Gaussian adjusted projected (GAP) measure of [23].

In B.2 we compute the 4-point function for this ensemble and show that the corresponding $\varepsilon_{nm}$ is approximately given by

$$
\varepsilon_{nm} = \frac{\hat{p}_n\hat{p}_m}{Z_1^2}\left(\frac{Z_2}{Z_1^2}-\frac{\hat{p}_n}{Z_1}-\frac{\hat{p}_m}{Z_1}\right)(1+\delta_{nm})+O\left(e^{-4S}\right)\,.
\tag{140}
$$

Notice that the scaling with the entropy is the same as (67). From this expression we can easily find the quantum deviation for the GAP ensemble:

$$
\Delta_O^2 = \frac{Z_2}{Z_1^2}\Big[\langle T_2|(\mathcal{O}_L-\langle\mathcal{O}_L\rangle_1)(\mathcal{O}_R-\langle\mathcal{O}_R\rangle_1)|T_2\rangle + O\left(e^{-S}\right)\Big]\,.
\tag{141}
$$

Here $\langle\mathcal{O}\rangle_1 = \langle T_1|\mathcal{O}|T_1\rangle$ and $|T_k\rangle$ was defined in (70). The expression above is not a connected two point function. To see this consider canonical probabilities: $|T_2\rangle$ is the thermofield double at temperature $2\beta$, while $\langle\mathcal{O}_L\rangle_1$, $\langle\mathcal{O}_R\rangle_1$ are thermal expectation values at temperature $\beta$. This means that the quantum deviation for the GAP ensemble doesn't have a holography dual as simple as the one for the ensembles we considered in sec. 4.

One more reason why we prefer the ensemble of sec. 4 over the GAP ensemble is the following. Consider the quantum deviation of the Hamiltonian,

$$
\Delta_H^2 = \frac{Z_2}{Z_1^2}\Big[\frac{1}{Z_2}\sum_n E_n^2\hat{p}_n^2 - \Big(\frac{1}{Z_2}\sum_n E_n\hat{p}_n^2\Big)^2\Big] + \frac{Z_2}{Z_1^2}\Big[\frac{1}{Z_2}\sum_n E_n\hat{p}_n^2 - \frac{1}{Z_1}\sum_n E_n\hat{p}_n\Big]^2\,.
\tag{142}
$$

The first term is the same as the one we found in sec. 4, up to the prefactor it is equal to the energy variance in the ensemble with probabilities $\hat{p}_n^2$. For canonical probabilities, it is the variance at temperature $2\beta$, which is of order $S$. The second term, again up to the prefactor, is

equal to the difference squared between the mean energy in the ensembles with probabilities $\hat{p}_n$ and $\hat{p}_n^2$. The second term is typically much bigger than the first. For example for canonical probabilities this is the difference squared between the thermal expectation values of the energy at temperature $\beta$ and $2\beta$, which is of order $S^2$. We conclude that the energy fluctuations in the GAP ensemble are much bigger than in the ensemble of sec. 4.

## B.1 Computing the 4-point function

The GAP measure is given by [23]

$$D\psi = \delta(1 - \|\psi\|)\Big(\prod_{l=1}^{d} \frac{d\psi_l^* d\psi_l}{2\pi i} \frac{Z_1}{\hat{p}_l}\Big)\int_0^\infty dr\, r^{2d+1} \exp\Big(-r^2 \sum_n \frac{Z_1}{\hat{p}_n}|\psi_n|^2\Big). \tag{143}$$

Here $d$ is the size of the Hilbert space. For the convenience of the reader we briefly review how this measure is constructed below, see B.2.

Before considering the 4-point function we check that this measure reproduces the correct two point function

$$\begin{aligned}
[\![\psi_n \psi_m^*]\!] &= \int \Big(\prod_{l=1}^{d} \frac{d\psi_l^* d\psi_l}{2\pi i} \frac{Z_1}{\hat{p}_l}\Big)\int_0^\infty dr\, r^{2d+1} \psi_n \psi_m^* \exp\Big(-r^2 \sum_n \frac{Z_1}{\hat{p}_n}|\psi_n|^2\Big)\delta(1 - \|\psi\|) \\
&= \int \Big(\prod_{l=1}^{d} \frac{d\Psi_l^* d\Psi_l}{2\pi i} \frac{Z_1}{\hat{p}_l}\Big)\Psi_n \Psi_m^* \exp\Big(-\sum_n \frac{Z_1}{\hat{p}_n}|\Psi_n|^2\Big) \\
&= \frac{\hat{p}_n}{Z_1}\delta_{mn}.
\end{aligned} \tag{144}$$

Going to the second line we have changed coordinates to $\Psi = r\psi$ and used the $\delta$ function to integrate over $r$; in the last line we have performed the Gaussian integrals.

Next we consider the 4-point function. Repeating the manipulations we have used for the 2-point function we find

$$[\![\psi_{n_1}\psi_{n_2}\psi_{m_1}^*\psi_{m_2}^*]\!] = \int \Big(\prod_{l=1}^{d} \frac{d\Psi_l^* d\Psi_l}{2\pi i} \frac{Z_1}{\hat{p}_l}\Big)\frac{\Psi_{n_1}\Psi_{n_2}\Psi_{m_1}^*\Psi_{m_2}^*}{\sum_n |\Psi_n|^2} \exp\Big(-\sum_n \frac{Z_1}{\hat{p}_n}|\Psi_n|^2\Big). \tag{145}$$

Compared to the 2-point function there is an extra factor $\sum_n |\Psi_n|^2$ in the denominator. To deal with this extra term we use the Schwinger trick $z^{-1} = \int_0^\infty \exp(-xz)dx$:

$$\begin{aligned}
[\![\psi_{n_1}\psi_{n_2}\psi_{m_1}^*\psi_{m_2}^*]\!] &= \int_0^\infty dx \int \Big(\prod_{l=1}^{d} \frac{d\Psi_l^* d\Psi_l}{2\pi i} \frac{Z_1}{\hat{p}_l}\Big)\Psi_{n_1}\Psi_{n_2}\Psi_{m_1}^*\Psi_{m_2}^* \\
&\quad \times \exp\Big[-\sum_n \Big(\frac{Z_1}{\hat{p}_n} + x\Big)|\Psi_n|^2\Big].
\end{aligned} \tag{146}$$

The integral over $\psi$ has factorized in a product of Gaussian integrals, it gives

$$\begin{aligned}
[\![\psi_{n_1}\psi_{n_2}\psi_{m_1}^*\psi_{m_2}^*]\!] &= \frac{\hat{p}_{n_1}\hat{p}_{n_1}}{Z_1^2}(\delta_{n_1 m_1}\delta_{n_2 m_2} + \delta_{n_1 m_2}\delta_{n_2 m_1}) \\
&\quad \times \int_0^\infty dx \Big(1 + \frac{\hat{p}_{n_1}}{Z_1}x\Big)^{-1}\Big(1 + \frac{\hat{p}_{n_2}}{Z_1}x\Big)^{-1}\prod_{l=1}^{d}\Big(1 + \frac{\hat{p}_l}{Z_1}x\Big)^{-1}.
\end{aligned} \tag{147}$$

We expect the integral in the second line, let's call it $I_{n_1 n_2}$, to be close to 1. We now estimate how close. First notice that the product in the square bracket is given by

$$\prod_{l=1}^{d}\Big(1+\frac{\hat{p}_l}{Z_1}x\Big)^{-1} = e^{-\operatorname{Tr}\log(1+\rho_1 x)}, \tag{148}$$

where $\rho_1 = 1/Z_1 \sum_n \hat{p}_n |n\rangle\langle n|$. The integrand in $I_{n_1 n_2}$ decays fast for $x > Z_1$, so we can restrict to smaller values of $x$ and expand the log in the equation above. To second order we have

$$
\begin{aligned}
e^{-\operatorname{Tr}\log(1+\rho_1 x)} &= \exp\Big[-\sum_{n=1}^{\infty}\frac{(-1)^{n-1}}{n}\frac{Z_n}{Z_1^n}x^n\Big] \\
&= \exp\Big[-x+\frac{Z_2}{2Z_1^2}x^2+O\Big(\frac{x^3}{d^3}\Big)\Big].
\end{aligned}
\tag{149}
$$

We perform the integral over $x$ by expanding the quadratic piece of the exponential and the two fractions left in (147), giving

$$I_{n_1 n_2} = 1 + \frac{Z_2}{Z_1^2} - \frac{\hat{p}_{n_1}}{Z_1} - \frac{\hat{p}_{n_1}}{Z_1} + O\Big(\frac{1}{d^2}\Big). \tag{150}$$

We find that the 4-point function is approximately given by

$$[\![|\psi_n|^2|\psi_m|^2]\!] = \frac{\hat{p}_n \hat{p}_m}{Z_1^2}\Big(1+\frac{Z_2}{Z_1^2}-\frac{\hat{p}_{n_1}}{Z_1}-\frac{\hat{p}_{n_1}}{Z_1}\Big)(1+\delta_{nm})+O\Big(\frac{1}{d^4}\Big). \tag{151}$$

If we subtract the Gaussian piece we find that indeed $\varepsilon_{nm}$ is given by (140).

## B.2 GAP overview

We briefly review the construction of the GAP measure, we refer to the original paper [23] for more details.

First notice that if we neglect the constraint $\langle\psi|\psi\rangle = 1$ we can generate the correct 2-point function with a simple Gaussian measure,

$$\mathcal{D}_{\mathrm{G}}\Psi = \Big(\prod_{l=1}^{d}\frac{d\Psi_l^* d\Psi_l}{2\pi i}\frac{Z_1}{\hat{p}_l}\Big)\exp\Big(-\sum_n \frac{Z_1}{\hat{p}_n}|\Psi_n|^2\Big). \tag{152}$$

Here to avoid confusion we will denote unnormalized states with $\Psi$.

Next we would like to restrict this measure to the set of normalized states without changing the two point function. The authors of [23] achieve this by first adjusting the Gaussian measure to

$$\mathcal{D}_{\mathrm{GA}}\Psi = \Big(\prod_{l=1}^{d}\frac{d\Psi_l^* d\Psi_l}{2\pi i}\frac{Z_1}{\hat{p}_l}\Big)\exp\Big(-\sum_n \frac{Z_1}{\hat{p}_n}|\Psi_n|^2\Big)\sum_m |\psi_m|^2. \tag{153}$$

It's easy to see that this measure is still properly normalized, $\int \mathcal{D}_{\mathrm{GA}}\Psi = \sum_n \hat{p}_n/Z_1 = 1$. Then they project on the unit sphere by integrating the Gaussian adjusted measure over the cone that starts at the origin and goes through an infinitesimal angle on the unit sphere. The final result is

$$\mathcal{D}_{\mathrm{GAP}}\psi = \delta(1-\|\psi\|)\prod_{l=1}^{d}\Big(\frac{d\psi_l^* d\psi_l}{2\pi i}\frac{Z_1}{\hat{p}_l}\Big)\int_0^{\infty}dr\, r^{2d+1}\exp\Big(-r^2\sum_n \frac{Z_1}{\hat{p}_n}|\psi_n|^2\Big). \tag{154}$$

Here $r$ is the radial coordinate along the cone and the $\delta$ function makes sure that $\psi$ lies on the unit sphere, $\|\psi\|^2 = \langle \psi | \psi \rangle = 1$.

We can check that for microcanonical probabilities, $\hat{p}_n = 1$ and $Z_1 = d$, this measures reduces to (131). The integrals over $r$ is

$$\int_0^\infty dr \, r^{2d+1} e^{-dr^2} = \frac{d!}{2d^{d+1}} \, . \tag{155}$$

Substituting this into the definition of $\mathcal{D}_{\text{GAP}}\psi$ we find again (131).

## C Factorizing ensemble

A particularly simple ensemble is given by pure states

$$|\phi\rangle = \frac{1}{\sqrt{Z_1}} \sum_{n=1}^{d} \sqrt{\hat{p}_n} e^{i\phi_n} |n\rangle \, , \tag{156}$$

with fixed magnitudes, $\hat{p}_n$, and i.i.d. random phases $\phi_n$, uniformly drawn from $[0, 2\pi)$. The average is given by an integral over these phases, for example

$$[\![ |\phi\rangle\langle\phi| ]\!] = \int \prod_n \frac{d\phi_n}{2\pi} |\phi\rangle\langle\phi| = \frac{1}{Z_1} \sum_n \hat{p}_n |n\rangle\langle n| \, , \tag{157}$$

and

$$[\![ |\phi\phi\rangle\langle\phi\phi| ]\!] = [\![ |\phi\rangle\langle\phi| ]\!] \otimes [\![ |\phi\rangle\langle\phi| ]\!] + \frac{1}{Z_1^2} \sum_{n \neq m} \hat{p}_n \hat{p}_m |n\,m\rangle\langle m\,n| \, . \tag{158}$$

One can check that for these ensembles the 4-point function factorizes[12]

$$[\![ |\phi_n|^2 |\phi_m|^2 ]\!] = [\![ |\phi_n|^2 ]\!] \, [\![ |\phi_m|^2 ]\!] \, . \tag{159}$$

The corresponding $\varepsilon_{nm}$ is

$$\varepsilon_{nm} = -\delta_{nm} \frac{\hat{p}_n \hat{p}_m}{Z_1^2} \, . \tag{160}$$

Notice that this is of the same order as the Gaussian piece, so this ensemble is not close to being Gaussian. Plugging this into (65) we discover that $\Delta_H^2 = 0$ for this ensemble. This has a simple explanation: the expectation value of the energy (or any other operator commuting with the Hamiltonian) only depends on the magnitudes $\hat{p}_n$, which are fixed in the ensemble.

Another way to see this is to notice that this ensemble can be obtained by time evolving one state from the ensemble with a sufficiently chaotic Hamiltonian. To be more explicit take as initial state the one where all the phases are zero, then the time evolved states are

$$|\psi(t)\rangle = \frac{1}{\sqrt{Z_1}} \sum_{n=1}^{d} \sqrt{\hat{p}_n} e^{-iE_n t} |n\rangle \, . \tag{161}$$

If the $\hat{p}_n$ are picked at sufficiently high energy, we can approximate the oscillating phases with random variables.

From what we've said above we see that this factorized ensemble is the natural one to consider if we are given one particular pure state which we then evolve for a time $t$. Even a small uncertainty on $t$ forces us to consider averages over the ensemble. However, the restriction to states with precisely the same expectation value for the energy seems unreasonable.

---

[12]In fact, this is the most general class of states obeying the factorization constraint. To see this, define the variable $x \equiv |\psi_n|^2 - [\![ |\psi_n|^2 ]\!]$. This is a real random variable. The factorization constraint is equivalent to $[\![ x^2 ]\!] = 0$ which implies $x = 0$. Therefore, the factorization constraint enforces that the amplitudes are fixed, $|\psi_n|^2 = q_n$.

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
