# Peer review of "Wormholes from Averaging over States"

_SciPost Physics, doi:SciPost Phys. 14, 026 (2023)_

## Round 1 · Referee Report · Anonymous (Referee 1) · 2021-10-20

Strengths

  1. Tackles an interesting subject.
  2. Provides interesting results.
  3. Well written.

Weaknesses

  1. No particular weaknesses to point out other than what I mention in the report.

Report

This paper studies the quantum deviation: a variance in the expectation value of operators over distributions of states. Such a quantity is of interest for several reasons. In the context of black hole physics, if one considers a family of states each dual to some black hole microstate which together describe the black hole in an ensemble sense, the quantum deviation can give a notion of variance of operator expectation values over the ensemble of states. Moreover, averaging over states has been discussed as a coarse-graining procedure which can explain the apparition of Euclidean (or spacetime) wormholes in the gravitational path integral.

After defining the quantum deviation and studying it in a simple one qubit model, the authors discuss how to pick an ensemble such that the quantum deviation can be computed using semi-classical gravity. The quantum deviation can also be rewritten as an averaged state but on a doubled Hilbert space. The authors show that by picking a suitable ensemble over states, the quantum deviation can be given by a L-R two-point function in the thermofield double state. The authors also go through a careful comparison of the canonical and microcanonical ensemble in this setting, including a Boltzmann-weighted microcanonical ensemble. Finally, the authors discuss the quantum deviation of the autocorrelation function $\langle O(t) O(0) \rangle $.

This is a well written paper that discusses an interesting topic which is the subject of much attention. I would be happy to recommend it for publication, but the authors need to clarify a few points, in particular point 1 in what follows.

Requested changes

  1. This is the most significant confusion I would like the authors to clear up: when one discusses the factorization puzzle and wormholes, it is important that we are discussing Euclidean or spacetime wormholes: these wormhole extrapolate between two-disconnected AdS boundaries. This should be distinguished from Lorentzian wormholes like the constant time-slice geometry of the thermofield double state (the Einstein-Rosen bridge). The full (Euclidean) geometry corresponds to a filling of a single AdS boundary. In that context, there is no factorization puzzle. The non-zero LR correlation function is simply due to entanglement. In the abstract and in the introduction, the authors are clearly referring to spacetime wormholes. But the result they obtain is a two-point function in the TFD state, which they claim is given by a wormhole calculation. But this is a very different type of wormhole! The authors should clearly distinguish the two, as the current formulation can be misleading for the reader.

  2. The rest are more technical comments. At the beginning of section 3.1, the authors talk about microcanonical vs canonical ensembles, and say that the thermodynamic limit is the $G_N\to0$ limit. There are two different thermodynamic limits in AdS/CFT. The standard thermodynamic limit: $G_N$ fixed but small, and $\beta\to0$. Another limit is to take $\beta$ fixed and send $G_N\to0$. This is not the standard thermodynamic limit and in such a case the two ensembles need not in general agree. My impression from the rest of the paper is that the authors consider mostly large AdS black holes and are thus in the standard thermodynamic limit, but the sentence mentioning $G_N\to0$ then confused me.

  3. Below (6.3), the authors discuss the decay of the thermal two-point function.This may be my ignorance, but I thought that the exponential damping eventually gets replaced by a polynomial decay. This was shown to be true for the spectral form factor, and also discussed for pure BH microstate in heavy-heavy-light-light computations. Does such a polynomial take-over not also happen for the thermal two-point function? Otherwise, the two-point function would be $e^{-e^{S}}$ at the dip time.

  4. When the authors study the quantum deviation of the two-point function, they restrict to Wick contractions for the four-point function. Is it clear that this is a legit approximation? At early times, it seems fine, but once the overall answer is already exponentially small in the entropy, should one really further distintinguish between polynomial corrections in $G_N$?

  • validity: top
  • significance: high
  • originality: high
  • clarity: top
  • formatting: perfect
  • grammar: perfect

Author:  Antonio Rotundo  on 2022-02-22  [id 2236]

(in reply to Report 1 on 2021-10-20)
Category:
remark
answer to question

We thank the referee for their valuable comments on the draft. We address them in order below.

  1. It is true that one does not need to invoke averages to make sense of the TFD state. However, our result shows that the wormhole connecting the two copies of the CFT can also be the result of an average over states. This is an explicit example of a connected geometry emerging from an average over states, and it strengthens the proposal that averages over states are sufficient to resolve the factorization problem.
    We agree with the referee that we should make clear that the wormhole we find is not a spacetime wormhole, as in usual discussions of the factorization puzzle. We've added a clarifying sentence about this point at the end of the introduction (last paragraph of introduction at pag 3).

  2. The referee is correct: the sentence mentioning $G_N\rightarrow 0$ is wrong. The point we were trying to make is that differences between the two ensembles can only be seen in terms subleading in the entropy (assuming the two ensembles are equivalent in the thermodynamic limit). In the bulk, these terms are invisible at leading order in $G_N$. The thermodynamic limit we use is indeed the standard one: $G_N$ fixed but small, and $β\rightarrow 0$.
    We have corrected this paragraph (second paragraph of section 3.1 at page 10).

  3. It's true that two-point functions of generic operators present late-time polynomial tails. However, we believe that to study information loss, it is more convenient to study operators which don't have such tails. This is true for the operators we have considered in section 6, i.e. primaries in a 2d CFT. In this case, the two-point function decays exponentially also at late times (until $t\sim S$). We haven't thought about which operators one should consider to avoid polynomial tails in higher dimensions. It would be interesting to address this question in future work.
    We have added a paragraph about this point to the conclusions (last bullet point at page 30).

  4. We use Wick contractions to calculate the 4-point function. This is never exponentially small in the entropy, but it is $O(1)$ in $G_N$ at all times. The exponential suppression in the quantum deviation is entirely due to the prefactor.
    We have added a sentence clarifying this point (above eq. 6.20 at page 27).

---

## Round 1 · Referee Report · Anonymous (Referee 2) · 2021-10-26

Strengths

  1. Addresses an interesting and timely subject - the role of ensemble averaging in holographic correspondence.
  2. Proposes concrete model of averaging with respect to ensemble of states, rather than ensemble of theories.

Weaknesses

  1. It is not clear if the proposed model/results actually resolve factorization paradox.

Report

The paper tries to quantify the idea that holographically, gravity "calculates" averaging over an ensemble of states in the given theory. The paper introduces the new quantity - quantum deviation - which essentially variance of expectation values, evaluated with respect to a certain ensemble of states. The authors discuss quantum deviation for different ensembles, and point out, for certain ensembles of states quantum deviation can be evaluated holographically, using classical gravity background (eternal black hole geometry). The discussion is focused on a particular observable: two-point function <A(t) A(0)>.

This is an interesting paper, which introduces interesting ideas . I recommend it for publication, after a suitable revision to clarify the issue explained below.

Requested changes

  1. The overall logic of the paper is not clear. The paper starts by discussing factorization paradox, stemming from an interpretation that gravity is dual to an ensemble (of theories). As a possible resolution the authors discuss if gravity can be dual to an ensemble of states (rather than theories). This, in a sense equivalent to a modification of the AdS/CFT dictionary, but the authors do not explain in detail what the new dictionary would be. Rather they proceed to discuss quantum deviation and notice (that for certain ensembles) quantum deviation can be calculated on the gravity side (quasi)-classically. Here they essentially use conventional AdS/CFT dictionary: they equate bulk calculation in the BH background with the expectation of some O_L x O_R in the particular thermofield double state (not an ensemble of states). In other words it is not entirely clear what the "rules of the game" are , if they are consistent, and if within this scenario factorization puzzle is in fact resolved. I would ask authors to clarify this moment.

  2. On page 14 equation (4.2) and the sentence immediately preceding it are confusing. It seems (4.2) is a result of simple calculation, but the sentence before it speaks of some "requirement."

  • validity: good
  • significance: high
  • originality: top
  • clarity: ok
  • formatting: perfect
  • grammar: perfect

Author:  Antonio Rotundo  on 2022-02-22  [id 2237]

(in reply to Report 2 on 2021-10-26)
Category:
remark
reply to objection

We thank the referee for their valuable comments on the draft. We address them in order below.

  1. We think that averaging over states does not constitute a modification of the AdS/CFT dictionary. More precisely, in AdS/CFT, every state in the CFT is associated to a state in the gravity side, and this is not changed. The reason why we consider averages over states is that, within the semiclassical approximation, gravity is not sensitive to the exact quantum state in the CFT, and only captures averages over ensembles of states. When one considers multiple copies of the theory, this averaging can lead to correlations between the different copies, as predicted by wormhole geometries.
    As usual, the eternal black hole corresponds to the thermofield double. What we are pointing out in the paper is that correlators in this state can also be interpreted as computing averages over ensembles of pure states. This is well known for the thermal ensemble, but, to our knowledge, not for the thermofield double.
    Finally, we don't claim that the factorization puzzle is resolved. However, we do think that averages over states are an interesting, and simple, candidate to find a resolution.

  2. The "requirement" is equation 4.1.
    We have changed this with an explicit reference to equation 4.1 (sentence right above eq. 4.2).

---

## Round 1 · Referee Report · Anonymous (Referee 3) · 2021-11-1

Strengths

  1. Provides timely insight on topic of great interest - paper addresses the topic of ensembles in understanding gravitational path integral

  2. Develops useful new tools/techniques - paper suggests 'quantum deviation' as an interesting diagnostic of ensembles of quantum states and suggests interesting ensembles to study

  3. Supports claims with careful and concrete calculations - paper gives careful analysis of deviation of various state ensembles and their correspondence with potential gravitational description

Weaknesses

No major weaknesses.

Report

Ensembles of quantum states have been proposed as a way to describe how gravity coarse-grains a quantum theory, and in particular to explain the existence of euclidean wormhole geometries. This paper investigates these claims and proposes the 'quantum deviation' as a useful probe of the variance of quantum expectations values in such an ensembled description.

The paper carefully analyzes the quantum deviation in a number of ensembles to assess their correspondence with a dual gravitational description. The paper also proposes interesting new ensembles of pure states to study.

These results are interesting and timely in an area of study that has captured a great deal of attention. The paper is well-written an generally quite clear. The calculations are carefully done and appear correct.

I recommend this paper for publication.

Requested changes

  1. Section 2.1 would benefit from some minor changes to improve clarity:

It is claimed that the deviation is sensitive to the average purity of states in the ensemble, but this claim can't be true in general. For example, the ensemble could be a single fixed density matrix that isn't pure, but the deviation would be zero. The text is not particularly clear about the approximation in Eq. (2.27) and under what assumptions it should hold.

The more general second term in (2.27) should contain the difference

$$ \mathrm{Tr}[[\rho^2]] - \mathrm{Tr}[[\rho]]^2
$$
rather than only the first term so that it is sensitive to the deviation of the purity rather than the purity itself. Of course, I think the authors are quite correct in the discussion around (2.25) that the purity of the average density matrix will often be much smaller than the average purity for the reasons they state (ie. the average is often more mixed than typical draws from the ensemble).

I think a few clarifying sentences here could alleviate some confusion.

  1. The notation used in Eq. (3.19) is unclear and introduces a function $g$ that hasn't been defined.

  • validity: high
  • significance: high
  • originality: high
  • clarity: good
  • formatting: perfect
  • grammar: perfect

Author:  Antonio Rotundo  on 2022-02-22  [id 2239]

(in reply to Report 3 on 2021-11-01)
Category:
remark
reply to objection

We thank the referee for their valuable comments on the draft. We address them in order below.

  1. It is true that the quantum deviation is not sensitive to the average purity in general. For example, to obtain eq. 2.27, we have assumed that $\sum_\rho P(\rho)^2\mathrm{Tr}[\rho^2]\ll\sum_\rho P(\rho)\mathrm{Tr}[\rho^2] $. This the reason why the term $−\mathrm{Tr}[[ρ]]^2$ is absent in eq. 2.27, and why the example provided by the referee is not described by eq. 2.27.
    We have slightly rearranged the discussion at the end of section 2.1 in a way that hopefully makes it less confusing.
    We don't agree that $\mathrm{Tr}[[ρ^2]]−\mathrm{Tr}[[ρ]]^2$ is the quantum deviation of the purity, which instead is $\mathrm{Tr}[[\rho^2\otimes \rho^2 ]]−(\mathrm{Tr}[[ρ^2]])^2$.

  2. This is a typo, in 3.19 there should be $h$, and not $g$. We have corrected this.

---

## Round 2 · List of Changes

• Added paragraph at the end of the introduction (page 3) to clarify that the wormhole we find is Lorentzian;
  • Corrected discussion of thermodynamic limit (second paragraph at beginning of sec. 3.1, page 10);
  • Added one bullet point about polynomial tails to the discussion (last paragraph of conclusion at page 30);
  • Added a sentence about higher order terms in G below eq. 6.19;
  • Explicit reference to eq. 4.1 (above eq. 4.2);
  • Rearranged discussion of purity (page 7 from eq. 2.23 to the end of sec. 2.1);
  • Replace g with h in eq. 3.19;
  • Swapped left and right basis states in eq. 3.6, 4.15, and 4.22;
  • Added Theta in eq. 4.14;
  • Changed notation for partial transpose in eq. 4.14, and 3.7;
  • Added a footnote at page 8 clarifying some aspects of the partial transpose.

---

## Editorial Decision

published